# *Salmonella*-induced SIRT1 and SIRT3 are crucial for maintaining the metabolic switch in bacteria and host for successful pathogenesis

Dipasree Hajra[1], Raju S Rajmani[2], Ayushi Devendrasingh Chaudhary[3,4], Shashi Kumar Gupta[3,4], Dipshikha Chakravortty[1,5]*

[1]Department of Microbiology and Cell Biology, Indian Institute of Science, Bangalore, India; [2]Centre of Infectious Disease Research, Indian Institute of Science, Bangalore, India; [3]Pharmacology Division, CSIR-Central Drug Research Institute, Lucknow, India; [4]Academy of Scientific and Innovative Research (AcSIR), Ghaziabad, India; [5]Adjunct Faculty, School of Biology, Indian Institute of Science Education and Research, Thiruvananthapuram, India

*For correspondence: dipa@iisc.ac.in

Competing interest: The authors declare that no competing interests exist.

## eLife Assessment

These authors present findings on the role of the sirtuins SIRT1 and SIRT3 during *Salmonella* Typhimurium infection. This **valuable** study increases our understanding of the mechanisms used by this pathogen to interact with its host and may have implications for other intracellular pathogens. The reviewers disagreed on the strength of the evidence to support the claims. Although one reviewer found the strength of the evidence **convincing**, the other found that it was **incomplete**, and that the main claims are only partially supported, as can be seen from the public reviews.

**Abstract** Sirtuins are the major players in host immunometabolic regulation. However, the role of sirtuins in the modulation of the immune metabolism pertaining to salmonellosis is largely unknown. Here, our investigation focussed on the role of two important sirtuins, SIRT1 and SIRT3, shedding light on their impact on intracellular *Salmonella*'s metabolic switch and pathogenesis establishment. Our study indicated the ability of the live *Salmonella* Typhimurium to differentially regulate the levels of SIRT1 and SIRT3 for maintaining the high glycolytic metabolism and low fatty acid metabolism in *Salmonella*. Perturbing SIRT1 or SIRT3 through knockdown or inhibition resulted in a remarkable shift in the host metabolism to low fatty acid oxidation and high glycolysis. This switch led to decreased proliferation of *Salmonella* in the macrophages. Further, *Salmonella*-induced higher levels of SIRT1 and SIRT3 led to a skewed polarization state of the macrophages from a pro-inflammatory M1 state toward an immunosuppressive M2, making it more conducive for the intracellular life of *Salmonella*. Alongside, governing immunological functions by modulating p65 NF-κB acetylation, SIRT1, and SIRT3 also skew *Salmonella*-induced host metabolic switch by regulating the acetylation status of HIF-1α and PDHA1. Interestingly, though knockdown of SIRT1/3 attenuated *Salmonella* proliferation in macrophages, in in vivo mice model of infection, inhibition or knockdown of SIRT1/3 led to more dissemination and higher organ burden, which can be attributed to enhanced ROS and IL-6 production. Our study hence reports for the first time that *Salmonella* modulates SIRT1/3 levels to maintain its own metabolism for successful pathogenesis.

## Introduction

Sirtuins are NAD+-dependent deacetylases that are present in all forms of life. Sirtuins comprise a conserved core catalytic domain that removes acetyl moiety from the lysine residues of proteins in the presence of NAD+ as a cofactor (*Landry et al., 2000*), giving rise to 2'O-acetyl-ADP-ribose and free nicotinamide as products (*Jackson and Denu, 2002*; *Sauve et al., 2001*). Free nicotinamide acts as a non-competitive inhibitor of sirtuins (*Bitterman et al., 2002*). They possess variable N terminal and C terminal domains that confer different subcellular localization, substrate specificity, and functions (*Sanders et al., 2010*). Mammals have seven sirtuins that are responsible for regulating various biological functions such as cell survival, apoptosis, oxidative stress, metabolism, and inflammation (*Lin and Fang, 2013*; *Guarente, 2007*). SIRT1, 6, and 7 have nuclear localization, SIRT2 is cytoplasmic, and SIRT 3, 4, and 5 comprise the mitochondrial SIRTs. In addition to their deacetylase activity, they possess ADP ribosylation (SIRT1, SIRT4, and SIRT6), desuccinylation and demalonylation (SIRT5), delipoylation (SIRT4), and demyristoylation and depalmitoylation (SIRT6) enzymatic activities (*Martínez-Redondo and Vaquero, 2013*). Previous studies have shown that SIRT1 gets activated in response to acute immune response and deacetylates RelA/p65 component of NFκB, thereby mediating its proteasomal degradation (*Liu et al., 2011*). On the other hand, it activates RelB component of NFκB pathway. RelB causes heterochromatinization of pro-inflammatory genes like *Tnfa* and *Il1b* (*El Gazzar et al., 2007*). SIRT1 activates peroxisome proliferator-activated receptor γ (PPARγ) coactivator-1α (PGC-1α), mediating a metabolic switch from glycolysis toward fatty acid oxidation (FAO). SIRT1-mediated RelB activation, in turn, activates SIRT3, causing the promotion of mitochondrial bioenergetics (*Liu et al., 2015b*). PGC-1α, a major player in mitochondrial biogenesis, activates SIRT3 (*Kong et al., 2010*), which in turn causes activation of PGC-1α, thereby fuelling a positive feedback loop. SIRT3 accounts for the major mitochondrial deacetylase, orchestrating several metabolic processes such as fatty acid oxidation (FAO), promotion of the Tricaroboxylic acid (TCA) cycle, and inhibition of Reactive oxygen species (ROS) production (*van de Ven et al., 2017*).

*Salmonella enterica* serovar Typhimurium is a facultative intracellular Gram-negative enteric pathogen, causing a wide array of infections ranging from self-limiting gastroenteritis to diarrhoea in humans (*Garai et al., 2012*). *S. enterica* serovar Typhi cause systemic infection in humans with typhoidal symptoms. Recent reports reported incidences of 21 million (*Bhutta, 2009*) typhoid cases and 93 million of non-typhoidal (*Majowicz et al., 2010*) cases round the year. The virulence of *Salmonella* is majorly regulated by two pathogenicity islands, namely, SPI-1 and SPI-2. It uses SPI-1 encoded T3SS and the effector proteins to invade host cells (*Haraga et al., 2008*). Inside the macrophages, they harbour within the *Salmonella* containing vacuoles (SCV) by virtue of its SPI-2 effectors (*Hajra et al., 2021*). Macrophages, dendritic cells, and neutrophils are responsible for successful dissemination throughout the body through the reticulo-endothelial system (RES) (*Haraga et al., 2008*).

Macrophages, serving as an intracellular niche for *Salmonella,* exhibit several continua of polarization states. At the two extreme ends of the spectrum lie the classically polarized M1 macrophages and alternatively activated M2 macrophages. M1 macrophages comprise of the proinflammatory antimicrobial state producing IL-1β, IL-6, TNF-α, IL-12, and IFN-γ cytokines and exhibit enhanced expression of CD80, CD86 surface markers. The anti-inflammatory M2 macrophages promote bacterial persistence by producing anti-inflammatory cytokines like IL-10 and TGF-β and show increased expression of Arg-1 and CD206 surface markers (*Lawrence and Natoli, 2011*; *Martinez et al., 2009*). To sustain the continuous production of proinflammatory cytokines, M1 macrophages rely on glycolysis for their energy requirements. On the other hand, M2 macrophages are fuelled by enhanced oxidative phosphorylation (OXPHOS) and FAO (*Namgaladze and Brüne, 2014*). It has been previously reported that sirtuins-mediated attuning of metabolism impacts polarization of macrophages in vivo. SIRT1 has the ability to promote the polarization of M2 macrophages and inhibit inflammation in macrophages of adipose tissue (*Hui et al., 2017*; *Jia et al., 2017*; *Kratz et al., 2014*). SIRT3 suppresses ROS by deacetylating and activating MnSOD (*Cimen et al., 2010*).

Several bacteria are known to subvert the host immune system toward an immunosuppressive state. *Salmonella* or *Mycobacterium* have evolved mechanisms to counteract the M1 state of the host macrophage. *Salmonella* Typhimurium uses its SPI-2 effectors to inhibit the recruitment of NADPH oxidase to the SCV, thereby preventing oxidative burst-mediated microbicidal activity (*Vazquez-Torres et al., 2000*). Similarly, *Mycobacterium bovis* bacillus Calmette–Guérin prevents NOS2 recruitment to phagosomes (*Miller et al., 2004*). *Salmonella* Dublin causes inhibition of the production of

pro-inflammatory cytokines like IL-18 and IL-12p70 (*Bost and Clements, 1997*). Moreover, *Myco-bacteria* inhibits NFκB signalling and IFN-γ-mediated downstream pathways (*Pathak et al., 2007*). Furthermore, *Yersinia enterocolitica* elicits an M2 response by inducing arginase-1 expression and TGF-β1 and IL-4 production (*Tumitan et al., 2007*). *Yersinia* TTSS effector LcrV induces an M2 pheno-type supposedly by IL-10 production (*Brubaker, 2003*).

Since SIRT1 and SIRT3 are the major modulators of the immunometabolic paradigm, we intend to decipher the role of SIRT1 and SIRT3 in influencing host and *Salmonella* metabolism. This study highlights the role of SIRT1 and SIRT3 in intracellular pathogen survival by promoting *Salmonella* glycolysis and concomitantly driving host metabolism toward FAO. Additionally, *Salmonella* trigger an immunosuppressive M2 environment conducive to its intravacuolar proliferation by modulating SIRT1 and SIRT3 levels. Here, we have shown that SIRT1 and SIRT3 knockdown cause decreased M2 surface marker expression such as CD206, along with increased production of pro-inflammatory cytokines and ROS generation, together amounting to attenuated bacterial intracellular proliferation within the infected macrophages. Moreover, SIRT1-mediated p65 NF-κB deacetylation played a vital role in immune function regulation within the *Salmonella*-infected macrophages with increased interaction of SIRT1 with p65 NF-κB. SIRT1 knockdown or inhibition resulted in hyperacetylation of p65 NF-κB, thereby leading to enhanced pro-inflammatory response in *S.* Typhimurium-infected macrophages. Further, SIRT1 and SIRT3 knockdown or inhibition skewed the *Salmonella*-induced host metabolic shift by regulating acetylation status of HIF-1α and PDHA. This caused increased host glycolysis and reduced FAO. However, the *Salmonella* shows the opposing metabolic profile with increased FAO and reduced glycolysis upon SIRT1 or SIRT3 inhibition. In contrast to the macrophages, in in vivo mice model of infection, SIRT1 and SIRT3 inhibition resulted in increased pathogen loads in organs and triggered enhanced bacterial dissemination, together leading to increased susceptibility of the mice to *S.* Typhimurium infection owing to increased ROS and IL-6 production. To the best of our knowledge, this is the first report implying the ability of host sirtuins in impacting intracellular bacterial metabolism crucial for successful pathogenesis.

## Results

### *Salmonella* modulates SIRT1 and SIRT3 expression along its course of infection in macrophages

Upon infection of RAW 264.7 murine macrophages with wildtype *S.* Typhimurium strain 14028S, we observed an increased expression level of *Sirt1* and *Sirt3* at initial and middle phases of infection, precisely at 2 hr and 6 hr post-infection through qPCR (*Figure 1A and B*). The *Sirt1* expression level declined at later phases of infection. On the other hand, the *Sirt3* transcript levels remained elevated at all time points with respect to uninfected control with a marked increment at 16 hr time point post-infection, which subsided at 20 hr post-infection. We even monitored the expression profile of *Sirt1* and *Sirt3* in primary macrophages like peritoneal macrophages of C57BL/6 mice and observed a similar trend of elevated expression at initial (2 hr), middle (6 hr), and late (16 hr) time points post-infection (*Figure 1C and D*). In confocal laser scanning microscopy (CSLM) studies, we observed a similar increase in SIRT1 and SIRT3 expression at 6 hr post-infection within the infected macrophages RAW 264.7 macrophages. (*Figure 1E–H*). Immunoblotting revealed increased protein expression of both SIRT1 and SIRT3 at 2 hr post-infection in comparison to the uninfected control (*Figure 1—figure supplement 1*). However, SIRT1 expression exhibits a gradual decline at the late phase of infection (*Figure 1—figure supplement 1A and C*). In line with the confocal microscopy data, SIRT3 immunoblotting data shows an increased protein expression profile at 6 hr and 16 hr post-infection (*Figure 1—figure supplement 1B–D*). Subsequently, to ascertain whether indeed *Salmonella* could modulate SIRT1 or SIRT3 expression levels, we evaluated the *Sirt1* and *Sirt3* mRNA and SIRT1 and SIRT3 protein expression profile within RAW 264.7 macrophages upon infection with wildtype *S.* Typhimurium and SPI-1 (Δ*invC*) (InvC, protein export apparatus) or SPI-2 (Δ*ssaV* and Δ*steE*) (SsaV, structural component of SPI-2 needle apparatus; SteE, SPI-2 effector protein involved in driving M2 polarization) mutants of *S.* Typhimurium. Our results depict the ability of wildtype *S.* Typhimurium to induce the expression of both SIRT1 and SIRT3 within the infected RAW 264.7 macrophages. However, infection with either SPI-1 or SPI-2 mutant abrogates the induction of *Sirt1,* whereas only SPI-2 mutants (Δ*ssaV* and Δ*steE*) and not SPI-1 mutant infection caused reduction in *Sirt3* transcript-level expression in the

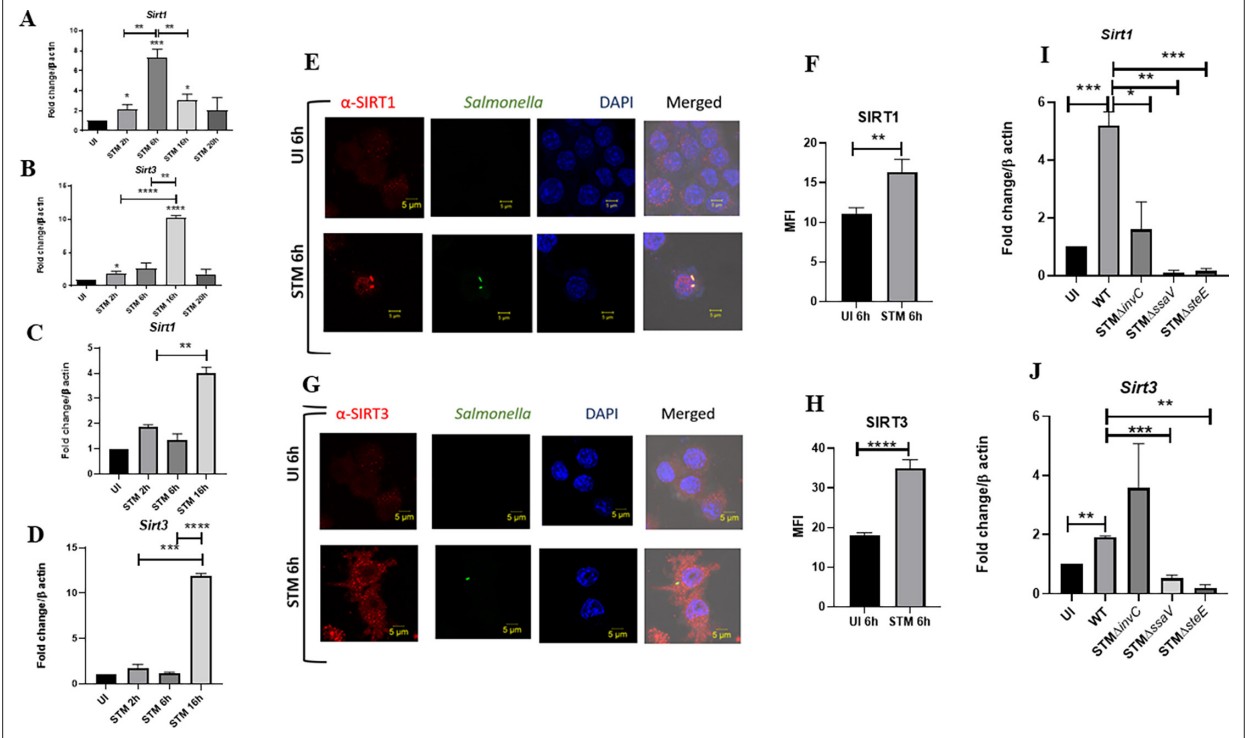

**Figure 1.** *Salmonella* modulates the expression of SIRT1 and SIRT3 along its course of infection. (**A, B**) Expression studies of *Sirt1* and *Sirt3* through qPCR in RAW 264.7 macrophages. Data is representative of N = 4, n = 2. Unpaired two-tailed Student's *t*-test was performed to obtain the p-values (****p<0.0001, ***p<0.001, **p<0.01, *p<0.05). (**C, D**) Expression studies of *Sirt1* and *Sirt3* through qPCR in peritoneal macrophages derived from C57BL/6. Data is representative of N = 3, n = 2. Unpaired two-tailed Student's *t*-test was performed to obtain the p-values (****p<0.0001, ***p<0.001, **p<0.01, *p<0.05). (**E**) Representative confocal images of RAW 264.7 macrophages exhibiting SIRT1 expression upon *S.* Typhimurium infection at indicated time points post-infection. Data is representative of N = 3, n = 80 (microscopic field). (**F**) Quantitative representation of the expression profile as depicted in the confocal images (**E**) in terms of mean fluorescence intensity (MFI). Unpaired two-tailed Student's *t*-test was performed to obtain the p-values (****p<0.0001, ***p<0.001, **p<0.01). (**G**) Representative confocal images of RAW 264.7 macrophages exhibiting SIRT3 expression upon *S.* Typhimurium infection at indicated time points post-infection. Data is representative of N = 3, n = 80 (microscopic field). (**H**) Quantitative representation of the expression profile as depicted in the confocal images (**G**) in terms of MFI. Unpaired two-tailed Student's *t*-test was performed to obtain the p-values (****p<0.0001, ***p<0.001, **p<0.01). (**I**) qPCR-mediated expression of *Sirt1* in RAW 264.7 macrophages upon infection with wildtype *S.* Typhimurium or SPI-1 (Δ*invC*)or SPI-2 (Δ*ssaV* and Δ*steE*) *m*utants of *S.* Typhimurium. Data is representative of N = 3,n = 3. Unpaired two-tailed Student's *t*-test was performed to obtain the p-values. (****p<0.0001, ***p<0.001, **p<0.01). (**J**) qPCR-mediated expression of *Sirt3* in RAW 264.7 macrophages upon infection with wildtype *S.* Typhimurium or SPI-1 (Δ*invC*) or SPI-2 (Δ*ssaV* and Δ*steE*) *m*utants of *S.* Typhimurium. Data is representative of N = 3, n = 3. Unpaired two-tailed Student's *t*-test was performed to obtain the p-values (****p<0.0001, ***p<0.001, **p<0.01).

The online version of this article includes the following source data and figure supplement(s) for figure 1:

**Figure supplement 1.** *Salmonella* modulates the expression of SIRT1 and SIRT3 along its course of infection.

**Figure supplement 1—source data 1.** Original files for western blot analysis displayed in *Figure 1—figure supplement 1A*.

**Figure supplement 1—source data 2.** Original files for western blot analysis displayed in *Figure 1—figure supplement 1B*.

**Figure supplement 1—source data 3.** File containing original uncropped western blots for *Figure 1—figure supplement 1A*, indicating relevant bands and treatments.

**Figure supplement 1—source data 4.** File containing original uncropped western blots for *Figure 1—figure supplement 1B*, indicating relevant bands and treatments.

infected macrophages, implicating the role of SPI-1 and SPI-2 genes in triggering SIRT1 and SIRT3 in the infected macrophages (*Figure 1I and J*). However, at the protein level, only SPI-2 mutant infection resulted in a predominant decline in SIRT3 expression and a mild reduction in SIRT1 expression (*Figure 1—figure supplement 1B*). Further, we examined the transcript-level profile of *Sirt1* and *Sirt3* in M1 or M2 polarized RAW 264.7 macrophages at 16 hr post-infection and observed 20-fold and 5-fold increase in *Sirt1* and *Sirt3* expression in M2-polarized-infected macrophages as opposed to 0.5-fold and 0.4-fold downregulation in M1-polarized-infected macrophages (*Figure 1—figure*

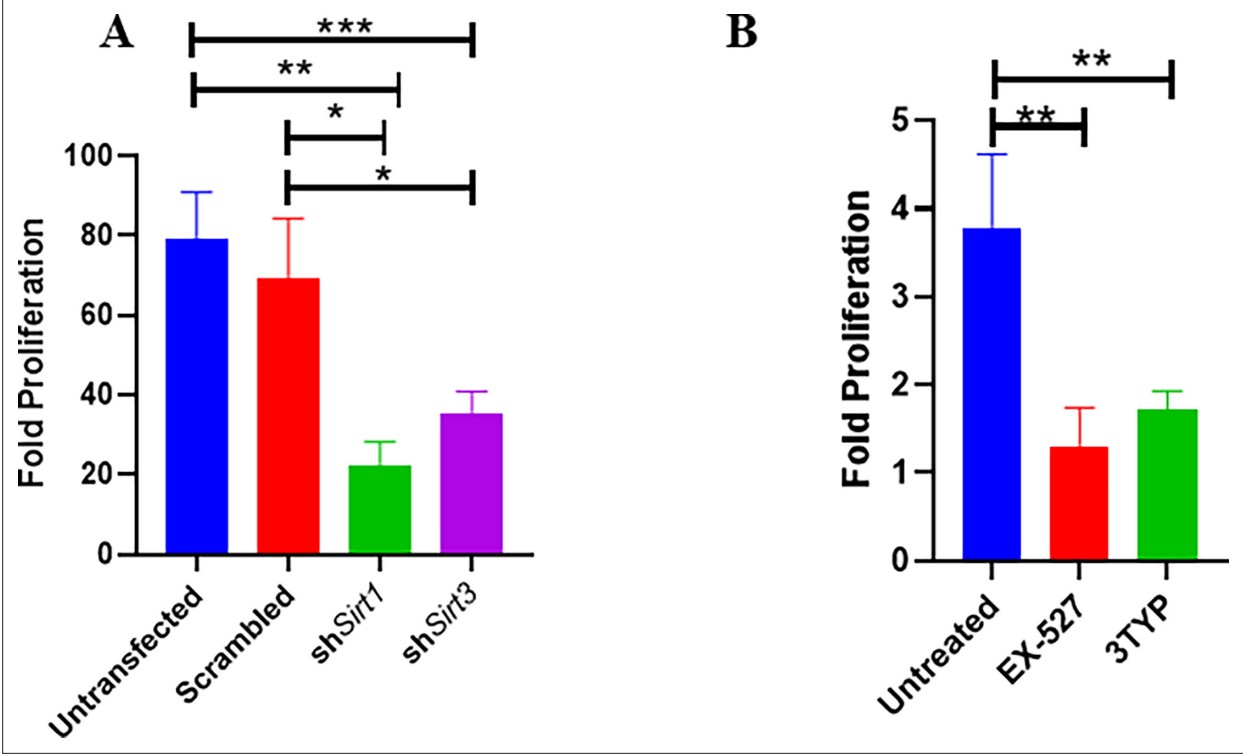

**Figure 2.** Effect of SIRT1 and SIRT3 knockdown in intracellular bacterial proliferation within RAW 264.7 and primary murine macrophages. (**A**) Fold proliferation of *S.* Typhimurium within RAW 264.7 macrophages in transfected and untransfected conditions. Data is representative of N = 3, n = 3. Unpaired two-tailed Student's *t*-test was performed to obtain the p-values (**p<0.01, *p<0.05). (**B**) Fold proliferation of *S.* Typhimurium within infected peritoneal macrophages isolated from adult male C57BL/6 mice upon SIRT1 (EX-527) or SIRT3 (3TYP) inhibitor treatment. Unpaired two-tailed Student's *t*-test was performed to obtain the p-values (**p<0.01, *p<0.05).

The online version of this article includes the following source data and figure supplement(s) for figure 2:

**Figure supplement 1.** Validation of *Sirt1* and *Sirt3* knockdown in RAW 264.7 macrophages.

**Figure supplement 1—source data 1.** Original files for western blot analysis displayed in *Figure 2—figure supplement 1C and D*.

**Figure supplement 1—source data 2.** File containing original uncropped western blots for *Figure 2—figure supplement 1C and D*, indicating relevant bands and treatments.

*supplement 1C and D*). Thus, an increase in expression profile both at transcript and protein levels indicates their role in *Salmonella* pathogenesis.

## SIRT1 and SIRT3 play a crucial role in intracellular bacterial proliferation in infected murine macrophages

As our previous *Sirt1* and *Sirt3* expression data in the polarized macrophages, hinted at the role of SIRT1 and SIRT3 in driving polarization of macrophages in the infected macrophages. We validated the intracellular replication of *S.* Typhimurium within the infected polarized macrophages. *S.* Typhimurium exhibited increased intracellular fold proliferation within anti-inflammatory M2-polarized macrophages in comparison to the pro-inflammatory M1-polarized RAW 264.7 macrophages (*Figure 1—figure supplement 1E and F*). To evaluate the role of SIRT1 and SIRT3 in the intracellular proliferation of the bacteria within murine macrophages, we have undertaken knockdown of *Sirt1* and *Sirt3* in RAW 264.7 (*Figure 2—figure supplement 1*) macrophages through PEI-mediated transfection of shRNA plasmids directed against *Sirt1* and *Sirt3*. Post 48 hr of transfection, the transfected cells were infected with multiplicity of infection, MOI = 10 of wildtype *S.* Typhimurium and a gentamicin protection assay was performed. Intracellular proliferation of the bacteria was quantified by plating the cell lysate at 2 hr and 16 hr post-infection. *Salmonella* exhibits compromised intracellular survival in *Sirt1* and *Sirt3* knockdown RAW 264.7 macrophages in comparison to the untransfected and scrambled controls (*Figure 2A*). Further, we have assessed the intracellular proliferation in

peritoneal macrophages isolated from thioglycolate-treated adult C57BL/6 mice post SIRT1 (EX-527) and SIRT3 (3TYP) inhibitor treatment. SIRT1 or SIRT3 inhibitor-treated macrophages exhibited attenuated intracellular replication in comparison to the untreated peritoneal macrophages (*Figure 2B*). Together, our results depict the role of SIRT1 and SIRT3 in controlling the intracellular proliferation of *S*. Typhimurium.

## SIRT1 and SIRT3 inhibition contributes to skewed inflammatory host responses upon *Salmonella* infection

Several reports indicate the role of SIRT1 and SIRT3 in the modulation of host immune responses pertaining to infection scenarios (*Elesela et al., 2020*; *Liu et al., 2015a*; *Yang et al., 2019*; *Kim et al., 2019*). Therefore, we intend to check whether SIRT1 or SIRT3 regulates immune functions in *Salmonella*-infected macrophages. To delineate the role of SIRT1 and SIRT3 in the modulation of immune responses, we wished to investigate the production of pro-inflammatory and anti-inflammatory cytokines in knockdown RAW 264.7 macrophages upon *S*. Typhimurium infection. Post 48 hr transfection, cells were subjected to wildtype *S*. Typhimurium infection at an MOI of 10. At the indicated time points, cell-free supernatant was harvested and evaluated for pro-inflammatory and anti-inflammatory cytokine production by ELISA. Inhibition of both SIRT1 and SIRT3 increased production of pro-inflammatory cytokine IL-6 significantly at 2 hr and 20 hr post-infection (*Figure 3—figure supplement 1A*). Moreover, there was only a significant reduction in anti-inflammatory IL-10 production upon *Sirt1* and *Sirt3* knockdown at 2 hr and 20 hr post-infection (*Figure 3—figure supplement 1B*). Further, we estimated the production of another pro-inflammatory cytokine, IL-1β at 20 hr post-infection under the knockdown condition of *Sirt1* and *Sirt3* (*Figure 3—figure supplement 1C*) and observed heightened IL-1β production under knockdown of *Sirt1* and *Sirt3* in comparison to the scrambled infected control. In peritoneal macrophages upon SIRT1 (EX-527-1 μM) or SIRT3 (3-TYP-1μM) chemical inhibitor treatment, an increase in IL-6 and IL-1β cytokine levels was observed at 6 hr post-infection (*Figure 3—figure supplement 1D and E*). This indicates the possible role of SIRT1 and SIRT3 in the regulation of cytokine production upon *Salmonella* infection.

Immune functions are an important determinant of macrophage polarization. Since SIRT1 and SIRT3 played an immunemodulatory role in *Salmonella* infection, we investigated whether *Salmonella* infection is associated with a shift in macrophage polarization status. To assess the ability of the pathogen to alter the polarization state of the macrophage, we have undertaken gene expression profiling of various M1 and M2 markers using nanoString nCounter technology along the course of *S*. Typhimurium infection at the indicated time points in RAW 264.7 macrophages. A gradual shift from pro-inflammatory M1 toward the anti-inflammatory M2 state was observed with the progression of *Salmonella* infection. Along the course of infection, there was a reduction in the expression of M1 markers like *Nos2*, *Cd40*,*Cd86*, *Tnfa*, *Nfkb2*, *Il6* and a corresponding increase in the expression of the M2 markers such as *Arg1*, *Ccl17*, *Cd206*, *IL4ra* with an exception of *Tgfb* (*Figure 3A*). In order to validate the polarization potency of the pathogen, FACS was performed using a pro-inflammatory M1 surface marker, CD86 tagged with PE. The data suggests a distinct decrease in CD86-positive population in the infected sets in comparison to the uninfected and the fixed dead bacteria control along the course of *S*. Typhimurium infection (*Figure 3B*, *Figure 3—figure supplement 2A*). Thus, the live pathogen has a propensity to skew the polarization state of the macrophage toward an anti-inflammatory M2 state to subvert the initial acute inflammatory response mounted by the host immune system.

To assess the role of SIRT1 and SIRT3 in macrophage polarization, we determined anti-inflammatory CD206 surface marker profiling of the infected macrophages through flow cytometry. Knockdown of *Sirt1* or *Sirt3* in infected RAW 264.7 macrophages resulted in a reduction in anti-inflammatory CD206 surface marker expression at 16 hr post-infection (*Figure 3C*, *Figure 3—figure supplement 2B*). Moreover, *Sirt1* or *Sirt3* knockdown led to enhanced intracellular ROS generation within the infected macrophages in comparison to the scrambled or the untransfected control (*Figure 3—figure supplement 3*). Further, the haematoxylin and eosin staining of the *S*. Typhimurium-infected mice liver tissue sections depicted exacerbated signs of inflammation in the SIRT1 (EX-527) or SIRT3 (3TYP) inhibitor-treated cohorts in comparison to the untreated controls with multiple necrotic foci and increased influx of inflammatory cell inflates. Moreover, these acute inflammatory signs of the liver sections get ameliorated in the SIRT1 (SRT1720) activator-treated infected cohort. Together, these data suggest the

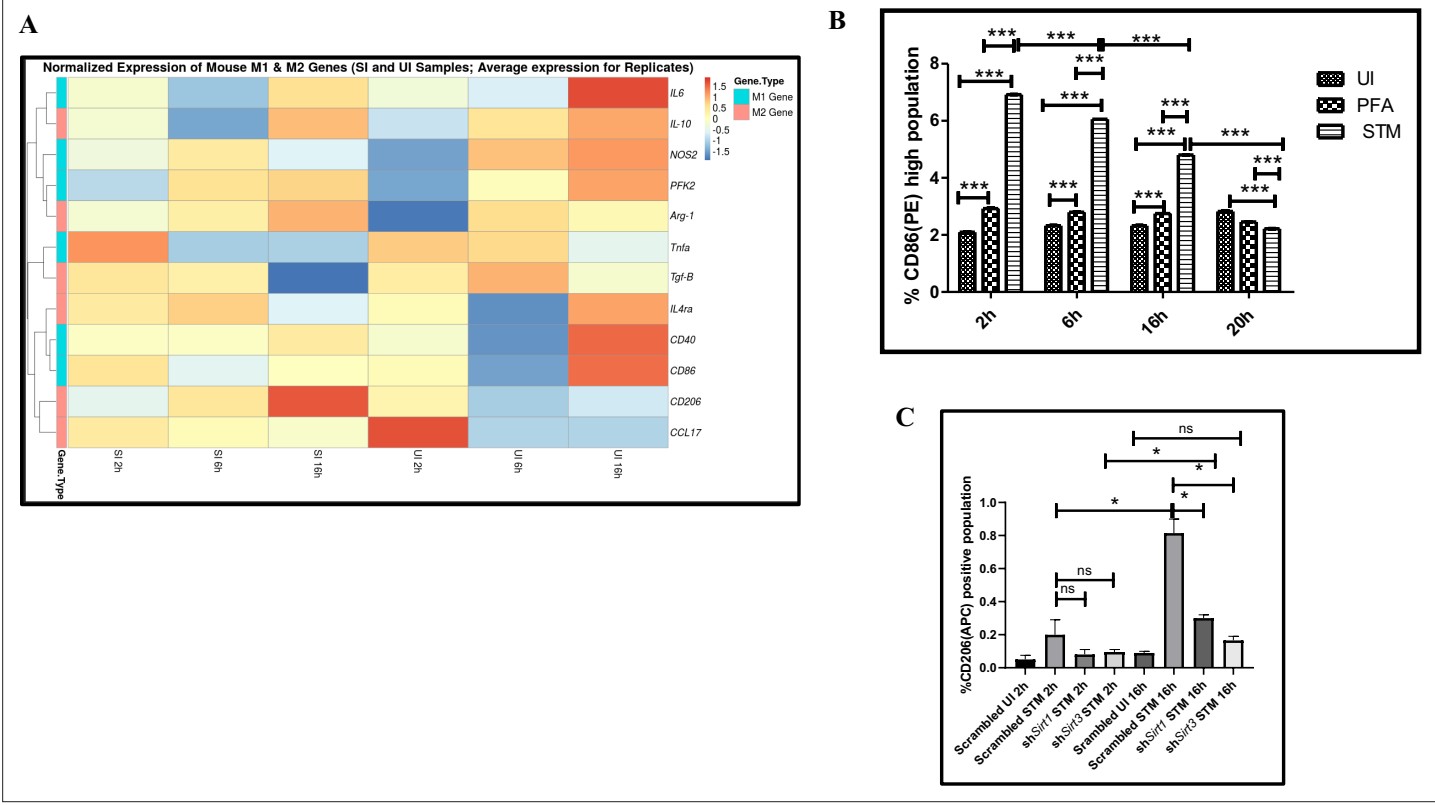

**Figure 3.** *Salmonella* Typhimurium skews the polarization state of the macrophage toward an immunosuppressive M2 state along the course of infection. (**A**) nanoString gene expression profiling data of *S.* Typhimurium (STM)-infected RAW 264.7 macrophages versus uninfected control data sets at 2 hr, 6 hr, and 16 hr time points of infection. Data is representative of N = 2, n = 2. (**B**) Quantitative representation of flow cytometric analysis of alteration in M1 CD86-positive population in STM-infected samples in comparison to uninfected (UI) and paraformaldehyde-fixed (PFA) bacteria at the indicated time post-infection. Data is representative of N = 2, n = 3. Two-way ANOVA and Bonferroni post-*t*-test were used to obtain p-values (***p<0.001). (**C**) Quantitative representation of flow cytometric analysis of M2 surface marker CD206 in STM-infected *Sirt1* or *Sirt3* knockdown RAW 264.7 macrophages in comparison to the scrambled control at the indicated time post-infection. Data is representative of N = 3, n = 3. Two-way ANOVA and Bonferroni post-*t*-test were used to obtain p-values (***p<0.001).

The online version of this article includes the following figure supplement(s) for figure 3:

**Figure supplement 1.** SIRT1 and SIRT3 modulate cytokine production in *Salmonella* infection scenario.

**Figure supplement 2.** *Salmonella* Typhimurium (STM) skews the polarization state of the macrophage toward an immunosuppressive M2 state along the course of infection.

**Figure supplement 3.** SIRT1 and SIRT3 play a role in maintaining ROS balance within *Salmonella*-infected macrophages.

**Figure supplement 4.** Effect of *Sirt1* and *Sirt3* knockdown or inhibition in intracellular bacterial proliferation within RAW 264.7 macrophages or peritoneal macrophages in the presence of ROS scavenger N-acetyl cysteine (NAC).

role of SIRT1 and SIRT3 in the modulation of host inflammatory response. Previous literature reports have shown that SIRT1 physically interacts with p65 subunit of NF-κB and inhibits transcription by deacetylating p65 at lysine 310 (*Yeung et al., 2004*). Moreover, SIRT1-mediated deacetylation of the p65 subunit of the master regulator of the inflammatory response, NF-κB, results in the reduction of the inflammatory responses mediated by this transcription factor (*Yang et al., 2012*). To evaluate the immune regulatory mechanism of SIRT1 in the *S.* Typhimurium (STM) infection scenario, we undertook SIRT1 immunoprecipitation in the infected RAW 264.7 macrophages at 16 hr post-infection alongside the uninfected macrophages and probed for NF-κB p65 interaction. We observed an increased inter-action of SIRT1 with NF-κB p65 in the infected macrophages in comparison to the uninfected control (*Figure 4A*). Further, the knockdown of *Sirt1* resulted in increased acetylation status of the NF-κB p65 upon infection in comparison to the scrambled, infected control (*Figure 4B and C*). To understand whether the enzymatic domain of SIRT1 possess any role in mediating this interaction, we carried out NF-κB p65 immunoprecipitation in infected RAW 264.7 macrophages in the presence or absence of

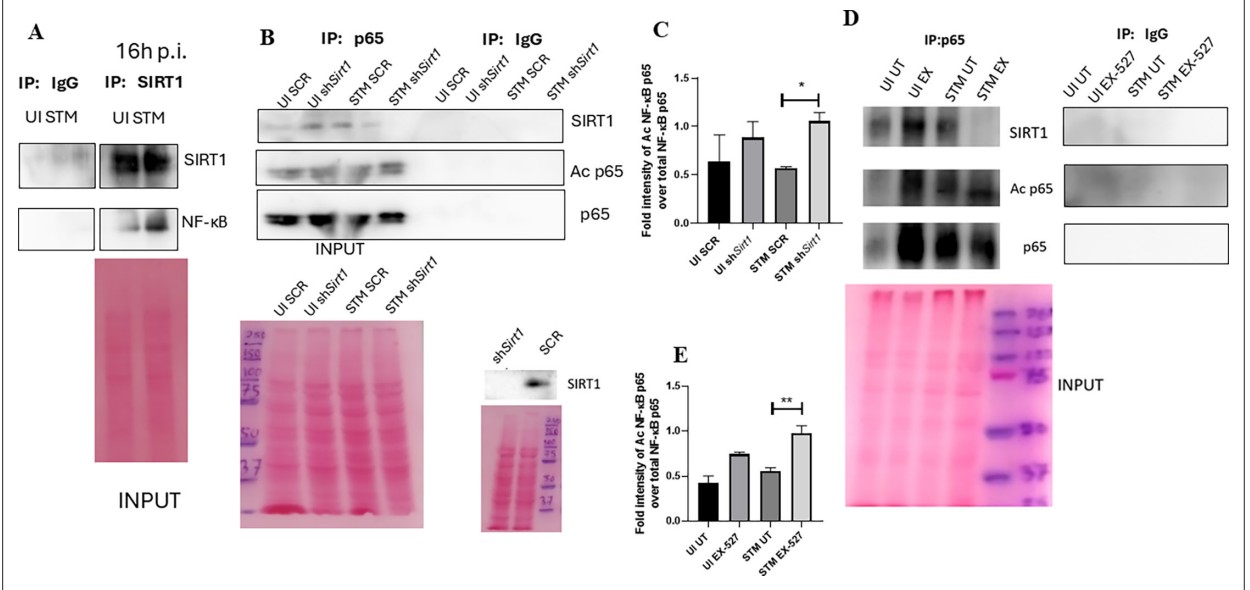

**Figure 4.** SIRT1 mediates modulation of immune functions via deacetylation of p65 subunit of NF-$\kappa$B in *S.* Typhimurium (STM)-infected macrophages. (**A**) Immunoblot depicting p65 NF-$\kappa$B interaction with SIRT1 post immunoprecipitation of SIRT1 in uninfected (UI) or STM-infected RAW 264.7 macrophages at 16 hr post-infection. Data is representative of N = 3, n = 1. (**B**) An immunoblot depicting p65 NF-$\kappa$B interaction with SIRT1 as well as the p65 NF-$\kappa$B acetylation status post immunoprecipitation of p65 (IP: p65) or with control IgG (IP: IgG) in UI or STM-infected RAW 264.7 macrophages upon knockdown with *Sirt1* shRNA or scrambled control. (**C**) Densitometric plot depicting the band intensities of acetylated p65 over total p65 in blot **B**. (**D**) An immunoblot depicting p65 NF-$\kappa$B interaction with SIRT1 as well as the p65 NF-$\kappa$B acetylation status post immunoprecipitation of p65 (IP: p65) or with control IgG (IP: IgG) in UI or STM-infected RAW 264.7 macrophages upon SIRT1 inhibitor (EX-527, 1 µM) treatment at 16 hr post-infection. UT, untreated. (**E**) Densitometric plot depicting the band intensities of acetylated p65 subunit of NF-$\kappa$B over total p65 NF-$\kappa$B in blot **D**.

The online version of this article includes the following source data for figure 4:

**Source data 1.** Original files for western blot analysis displayed in *Figure 4A*.

**Source data 2.** Original files for western blot analysis displayed in *Figure 4B*.

**Source data 3.** Original files for western blot analysis displayed in *Figure 4D*.

**Source data 4.** File containing original uncropped western blots for *Figure 4A*, indicating relevant bands and treatments.

**Source data 5.** File containing original uncropped western blots for *Figure 4B*, indicating relevant bands and treatments.

**Source data 6.** File containing original uncropped western blots for *Figure 4D*, indicating relevant bands and treatments.

SIRT1 catalytic chemical inhibitor, EX-527 (1 µM) treatment at 16 hr post-infection. We observed an increased interaction of NF-κB p65 with SIRT1 in the infected untreated macrophages compared to the untreated uninfected control (*Figure 4D*). However, the interaction of NF-κB p65 with SIRT1 gets abrogated under the EX-527 inhibitor treatment in the infected macrophages, thereby implying the function of the catalytic domain in mediating the interaction (*Figure 4D*). Moreover, an increased acetylation status of NF-κB p65 was observed in the EX-527-treated infected macrophages in comparison to the untreated infected macrophages (*Figure 4E*).

## SIRT1 and SIRT3 relieve oxidative stress in infected macrophages, and alleviation of the intracellular ROS generation restores intracellular survival of *S.* Typhimurium within the SIRT1 or SIRT3 knockdown macrophages

Previous reports have suggested the role of SIRT1 and SIRT3 in oxidative stress conditions. They are known to act in concert as antioxidants by reducing ROS production (*Singh et al., 2018*; *Kitada et al., 2019*; *Merksamer et al., 2013*). Moreover, enhanced ROS production is also a prototype of the classically activated macrophages (*Rendra et al., 2019*; *Zhou et al., 2018*). Here, we examined the effect of *Sirt1* and *Sirt3* knockdown in intracellular ROS generation in *S.* Typhimurium-infected RAW 264.7 macrophages through 2',7'-dichloro fluorescein diacetate (DCFDA) staining in FACS. Results depicted

significant enhancement in the production of ROS at 16 hr post-infection upon knockdown of *Sirt1* or *Sirt3* in comparison to untransfected or scrambled control (*Figure 3—figure supplement 3A and B*). Upon detection of extracellular ROS generation through phenol red hydrogen peroxidase assay, we detected higher ROS generation upon *Sirt3* KD at 6 hr post-infection and greater ROS production at 16 hr and 20 hr time points in *Sirt1* knockdown macrophages (*Figure 3—figure supplement 3C*). Therefore, SIRT1 and SIRT3 play an important role in mitigating the oxidative burst in *Salmonella*-infected macrophages. Our previous findings depicted decreased intracellular burden of *S.* Typhimurium within the *Sirt1* or *Sirt3* knockdown macrophages along with increase in intracellular ROS generation. Therefore, we speculated that the decreased intracellular proliferation within the *Sirt1* or *Sirt3* knockdown macrophages might be on account of increased intracellular ROS production, which might lead to increased killing of the intracellular bacteria. This hypothesis led us to evaluate the intracellular bacterial burden within the infected *Sirt1* or *Sirt3* knockdown RAW 264.7 macrophages or SIRT1 or SIRT3 inhibitor-treated peritoneal macrophages (isolated from C57BL/6 adult mice post fifth day of thioglycolate injection) in the presence of a ROS inhibitor named N-acetyl cysteine (NAC). NAC acts as a scavenger of ROS by antagonizing the activity of proteasome inhibitors (*Halasi et al., 2013*). The attenuated intracellular proliferation of *S.* Typhimurium within the knockdown or the chemical inhibitor-treated macrophages got restored upon 1 mM treatment of ROS scavenger, NAC (*Figure 3—figure supplement 4*). Therefore, intracellular ROS production within the knockdown murine macrophages is one of the reasons for the attenuated survival of the bacteria.

## SIRT1 and SIRT3 play a crucial role in mediating metabolic switch in infected macrophages

Macrophage polarization is not only governed by immunological changes but also contributed by metabolic reprogramming (*De Santa et al., 2019*; *Kratz et al., 2014*; *Galván-Peña and O'Neill, 2014*). Since previous data suggested progression of *Salmonella* infection with the shift in polarization state of the macrophage, we decided to investigate alteration of the metabolic state of the macrophages as macrophage polarization is governed by immune-metabolic shift. In pursuit of fulfilling such requirement, we performed gene expression studies of various metabolic genes through nanoString nCounter technology in *S.* Typhimurium-infected RAW 264.7 macrophages. Analysis of the gene profile revealed upregulation of genes involved in FAO and tricarboxylic acid cycle and corresponding downregulation of genes involved in glycolysis (*Figure 5A*). To validate the findings, we carried out qRT PCR to quantitatively measure the expression of a FAO gene, *Pppard*, in infected RAW 264.7 macrophages. We found that the mRNA level was elevated to twofold at 2 hr and 6 hr post-infection. At the late phase of infection, 16 hr post-infection around sixfold upregulation in mRNA transcript was noted (*Figure 5B*). Lactate estimation assay of *S.* Typhimurium-infected RAW 264.7 macrophages at the initial time point of 2 hr and at the late time point of 16 hr post-infection revealed a decline in lactate (glycolysis end product) production at 16 hr in comparison to 2 hr post-infection time point (*Figure 5C*). Together, these results suggest the capability of the pathogen to drive a shift in the metabolic status of the infected macrophages toward FAO. Next, we evaluated the function of SIRT1 and SIRT3 in influencing the metabolic switch in the infected macrophages through qRT PCR with several host fatty acid oxidizing genes (*Acox1*, *Hadha*, *Pdha1*, and *Acadl*) and glycolytic gene (*Pfkl*) in SIRT1 and SIRT3 knockdown macrophages (*Figure 5—figure supplement 1A*) and via lactate production assay (*Figure 5D and E*). Lactate estimation assay in *Sirt1* and *Sirt3* knockdown condition revealed enhanced lactate production at 16 hr post-infection in comparison to the scrambled control, which further authenticates the increased host glycolysis upon *Sirt1* and *Sirt3* knockdown scenario (*Figure 5D and E*). *Sirt1* and *Sirt3* knockdown RAW 264.7-infected macrophages showed decreased expression of fatty acid oxidizing genes and increased expression of glycolytic *Pfkl* gene in comparison to the scrambled infected control (*Figure 5—figure supplement 1A*). Similar results were obtained in the infected peritoneal macrophages under the SIRT1 or SIRT3 catalytic inhibitor treatment (*Figure 5—figure supplement 1B–D*). Moreover, qRT PCR-mediated metabolic gene profiling of liver and spleen isolated from wildtype *S.* Typhimurium-infected macrophages revealed decreased transcript-level expression of murine FAO genes upon treatment with SIRT1 (EX-527) or SIRT3 (3TYP) catalytic inhibitors, which got reversed upon SIRT1 activator (SRT1720) treatment (*Figure 5—figure supplement 1E and F*). Moreover, SIRT1 and SIRT3 knockdown or catalytic inhibition in peritoneal macrophages resulted in increased protein expression of host glycolytic genes such as phosphoglycerate kinase

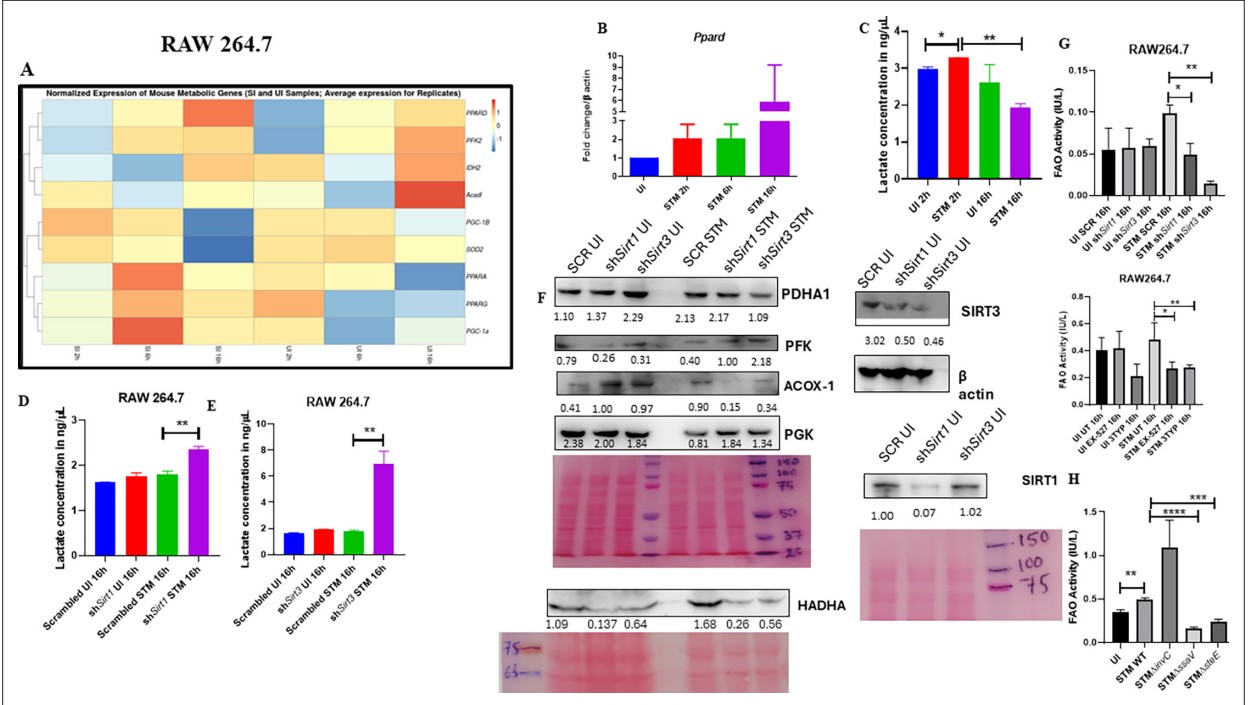

**Figure 5.** *Salmonella* Typhimurium drives the metabolism of the infected macrophage toward fatty acid oxidation. (**A**) Metabolic gene expression data of *S.* Typhimurium-infected RAW 264.7 macrophages at 2 hr, 6 hr, and 16 hr time points of infection through nanoString. Data is representative of N = 2, n = 1. (**B**) *Ppard* qPCR expression data in *S.* Typhimurium-infected RAW 264.7 macrophages at the indicated time points of infection. Data is representative of N = 3, n = 2. (**C**) Lactate estimation assay of *S.* Typhimurium-infected RAW 264.7 macrophages at the initial time point of 2 hr and at the late time point of 16 hr post-infection. Data is representative of N = 3, n = 3. Unpaired two-tailed Student's *t*-test was performed to obtain the p-values (**p<0.01, *p<0.05). (**D, E**) Lactate estimation assay of *S.* Typhimurium-infected RAW 264.7 macrophages upon *Sirt1* (**D**) or *Sirt3* (**E**) knockdown condition at 16 hr post-infection. Data is representative of N = 3, n = 3. Unpaired two-tailed Student's *t*-test was performed to obtain the p-values. (**F**) Immunoblotting of host glycolytic (PGK), TCA cycle (PDHA1), and fatty acid oxidation (HADHA, ACOX-1) proteins under *Sirt1* and *Sirt3* knockdown condition of *S.* Typhimurium-infected RAW 264.7 macrophages at 16 hr post-infection. (**G**) Fatty acid oxidation (FAO) assay of uninfected (UI) and infected (STM) RAW 264.7 macrophages under *Sirt1* or *Sirt3* knockdown or inhibitor treatment. N = 2, n = 2 (** p<0.01, *p<0.05). (**H**) FAO assay of uninfected (UI) and infected RAW 264.7 macrophages under infection with wildtype *S.* Typhimurium (STM WT), SPI-1 (ΔinvC), or SPI-2 (ΔssaV and ΔsteE) mutants of *S.* Typhimurium. Data is representative of N = 2,n = 2 (**p<0.01, * p<0.05).

The online version of this article includes the following source data and figure supplement(s) for figure 5:

**Source data 1.** Original files for western blot analysis displayed in *Figure 5F*.

**Source data 2.** File containing original uncropped western blots for *Figure 5F*, indicating relevant bands and treatments.

**Figure supplement 1.** SIRT1 and SIRT3 knockdown or inhibitor treatment led to decrease in fatty acid oxidation and promoted host glycolysis in *S.* Typhimurium-infected peritoneal macrophages.

**Figure supplement 1—source data 1.** Original files for western blot analysis displayed in *Figure 5—figure supplement 1D*.

**Figure supplement 1—source data 2.** File containing original uncropped western blots for *Figure 5—figure supplement 1D*, indicating relevant bands and treatments.

(Pgk), phosphofructokinase (Pfk) with concomitant reduction in protein expression of TCA cycle gene like pyruvate dehydrogenase (Pdha1) and FAO genes such as Hadha and Acox1 (*Figure 5F*, *Figure 5—figure supplement 1D*). FAO assay in the RAW 264.7 macrophages under *Sirt1* or *Sirt3* knockdown condition or inhibition treatment revealed a significant decrease in fatty acid β oxidation activity of the infected macrophages in comparison to the scrambled or the untreated control (*Figure 5G*). To investigate whether the host metabolic switch toward increased FAO is being driven by the pathogen, we performed fatty acid β oxidation activity assay under wildtype *S.* Typhimurium (STM WT), SPI-1 (ΔinvC), or SPI-2 (ΔssaV and ΔsteE) mutants of *S.* Typhimurium infection condition. We found that the wildtype bacteria with its intact SPI-2 effector secretion apparatus could promote increased host fatty acid β oxidation. In contrast, the SPI-2 (ΔssaV and ΔsteE) mutants of *S.* Typhimurium failed to drive host metabolic shift toward increased FAO (*Figure 5H*).

Collectively, these data suggest the role of SIRT1 and SIRT3 in mediating the *Salmonella*-induced host metabolic shift in the infected macrophages.

## SIRT1 and SIRT3 concomitantly influence *Salmonella* metabolism

Our previous data indicated a shift in host metabolism toward increased FAO along the course of *S*. Typhimurium infection in murine RAW 264.7 macrophages. *S*. Typhimurium drives the metabolism of the infected macrophage toward FAO. This observation led us to investigate the influence of host metabolic shift on the metabolic status of the pathogen harbouring inside the infected macrophages (*Taylor and Winter, 2020*). We were intrigued whether increased glucose availability within the fatty acid oxidizing macrophages is utilized by the bacteria. Thus, we undertook simultaneous gene expression studies of various *Salmonella* genes involved in their pathogenesis and metabolism through nanoString nCounter technology in *S*. Typhimurium-infected RAW 264.7 macrophages. The nanoString gene profile revealed enhanced expression of genes involved in *Salmonella* glycolysis and glucose uptake such as *pfkA* and *ptsG*, respectively (*Figure 6A*). This finding indicates the ability of the pathogen to drive the metabolic state of the host toward FAO with corresponding increased glucose utilization by the bacteria favouring their survival inside the host. qRT PCR results with several bacterial fatty acid oxidizing genes (*fadA, fadB, fadL, aceA, aceB*) and glycolytic genes (*ptsG*) in knockdown condition of *Sirt1* and *Sirt3* in RAW 264.7 macrophages further validated the nanoString gene expression profiles (*Figure 6B*). In scrambled control, *Salmonella* infection progresses with increased *Salmonella* glycolysis and reduced bacterial FAO. However, knockdown of *Sirt1* and *Sirt3* abrogates this bacterial metabolic shift by reducing its glycolysis and exhibiting enhanced FAO, thereby attenuating pathogen intracellular survival. Similar observations were obtained from the qPCR data in the infected mice liver and spleen samples with increased transcript-level expression of bacterial FAO genes and decreased expression of bacterial glycolytic genes upon SIRT1 or SIRT3 inhibitor treatment (*Figure 6C and D*). Therefore, SIRT1 and SIRT3-driven host metabolic switch potentially influences the metabolic profile of the intracellular pathogen.

## Mechanism behind SIRT1 or SIRT3-mediated metabolic switch

As per our previous findings, SIRT1 or SIRT3 inhibition led to increased host glycolysis and decline in fatty oxidation in the infected macrophages. HIF1α is a master regulator of glycolysis in host during stress conditions (*Kierans and Taylor, 2021*). Previous reports have suggested HIF1α to be a target of deacetylation by SIRT1 at Lys 674, which contribute to metabolic reprogramming in cancer cells. During hypoxia, downregulation of SIRT1 leads to increased acetylation and activation of HIF1α (*Lim et al., 2010*). Additionally, in CD4+ T cells, ectopic expression of SIRT1 inhibited IL-9 production and glycolysis by negatively regulating HIF1α (*Wang et al., 2016*). To delve into the mechanism behind SIRT1-mediated modulation of metabolic responses, we assessed the interaction of SIRT1 with HIF-1α in infected RAW 264.7 macrophages. The immunoprecipitation studies of SIRT1 showed increased interaction of the SIRT1 with HIF1α in the *S*. Typhimurium infection scenario with respect to the uninfected control (*Figure 7A*). Further, we evaluated the acetylation status of HIF1α in the *Sirt1* knockdown status of the infected macrophages. We found that SIRT1 knockdown showed increased acetylation of HIF1α in the infected macrophages in comparison to the scrambled infected control at 16 hr post-infection (*Figure 7B and C*). Immunoprecipitation studies under SIRT1 (EX-527) inhibitor treatment in RAW 264.7 macrophages revealed increased acetylation of HIF1-α along with reduced interaction of HIF-1α with SIRT1, thereby indicating the probable role of the catalytic domain in influencing the interaction (*Figure 7D–F*). Further, to ascertain the role of HIF-1α in mediating the glycolytic switch in the infected macrophages, we estimated the lactate production under *Sirt1* and *Sirt3* knockdown conditions in the presence or absence of HIF-1α inhibitor (chetomin) (*Viziteu et al., 2016*). Our results depicted a decline in lactate production upon chetomine treatment, including under *Sirt1* and *Sirt3* knockdown conditions (*Figure 7G*). Together, our results implicate the role of SIRT1 in governing glycolytic shift in the infected macrophages by deacetylating HIF1α. Upon SIRT1 knockdown or inhibition, HIF1α gets hyperacetylated, which causes activation of the downstream glycolytic genes. Alternatively, several key literature suggest the role of SIRT3 in modulating metabolic programming by deacetylating several proteins involved in FAO, the tricarboxylic acid cycle, and OXPHOS (*Hirschey et al., 2010*; *Zhang et al., 2021*). PDHA1 (pyruvate dehydrogenase E1 subunit

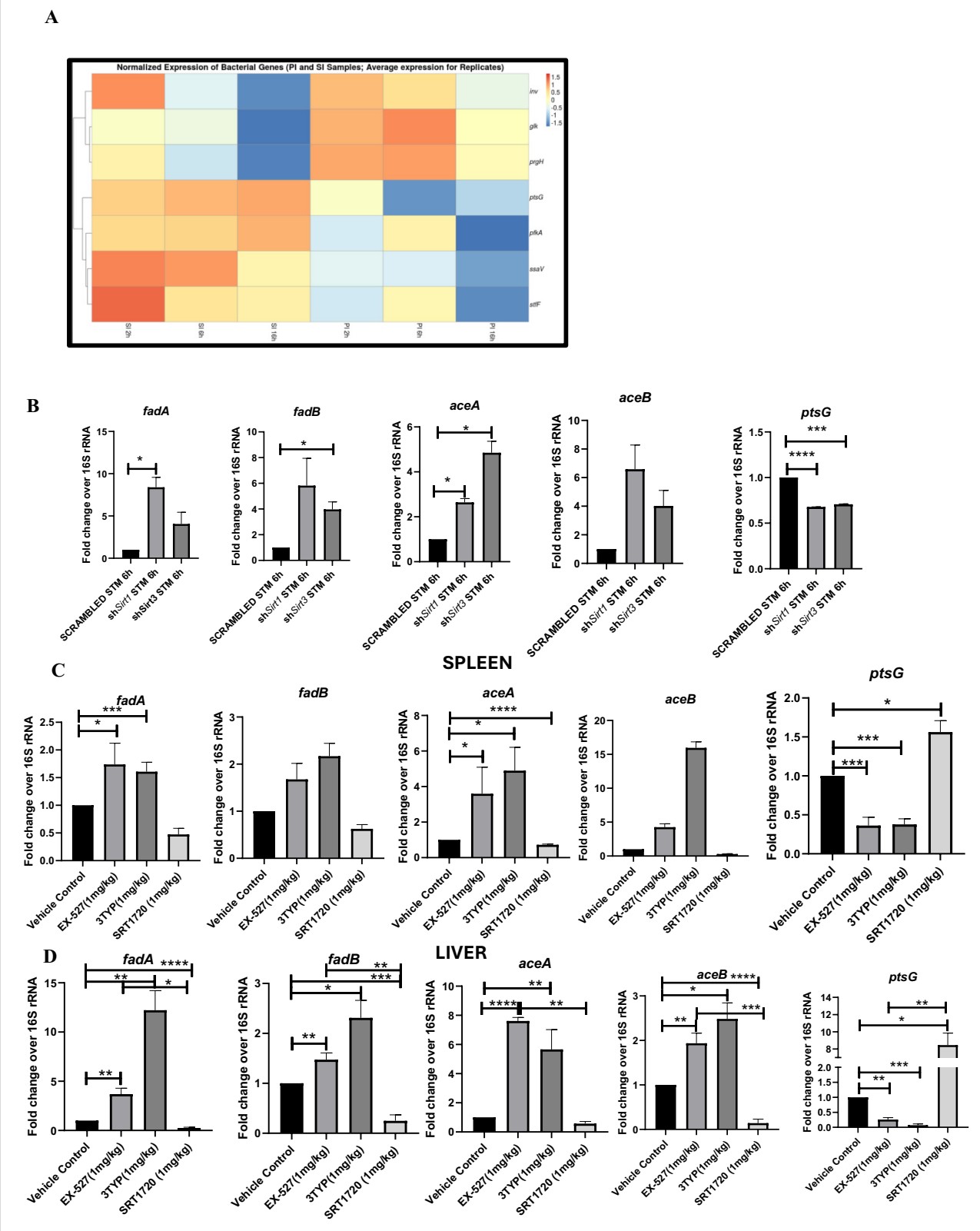

**Figure 6.** *Salmonella* Typhimurium infection proceeds with increased glycolysis and glucose uptake inside the infected RAW 264.7 macrophages. (**A**) *Salmonella* gene expression profiling data of *S.* Typhimurium-infected RAW 264.7 macrophages at 2 hr, 6 hr, and 16 hr time points of infection through nanoString. SI, *S.* Typhimurium-infected; PI, PFA fixed *S.* Typhimurium-infected. SPI-1 genes, *inv, prgH*; SPI-2 genes, *ssaV,stfF*; *glk*, glucokinase; *pfkA*, phosphofructose kinase A; *ptsG*, phosphophenolpyruvate-dependent sugar phosphotransferase system (PTS). Data is representative of N =

*Figure 6 continued on next page*

*Figure 6 continued*

2, n = 2. (**B**) qRT PCR gene expression profiling of *Salmonella* metabolic genes within infected RAW 264.7 macrophages in knockdown condition of either *Sirt1* or *Sirt3* at 6 hr post-infection. Data is representative of N = 3, n = 3. (**C, D**). qRT PCR gene expression profiling of *Salmonella* metabolic genes *within* infected female C57BL/6 mice spleen (**C**) and liver (**D**) under SIRT1 or SIRT3 inhibitor treatment harvested on fifth day post-infection of *S.* Typhimurium ($10^7$ CFU units/animal). Unpaired two-tailed Student's *t*-test was performed to obtain the p-values. Data is representative of N = 3, n = 3 (****$p<0.0001$, ***$p<0.001$, **$p<0.01$, *$p<0.05$).

alpha) is a key enzyme linking glycolysis to TCA cycle and OXPHOS. SIRT3 regulates PDHA1 acetylation by deacetylating PDHA1 at lysine 385 residue, thereby playing a key role in metabolic reprogramming (*Zhang et al., 2021*). PDHA1 acetylation coincides with PDH activity and increased PDHA1 phosphorylation (*Zhang et al., 2021*). Therefore, we investigated the role of SIRT3 in the modulation of host FAO upon *S.* Typhimurium infection in RAW 264.7 macrophages. To do so, we immunoprecipitated PDHA1 and checked for its interaction with SIRT3 or SIRT1 under the knockdown condition of SIRT3 or upon SIRT3 inhibitor treatment (*Figure 7H, I and J*). We observed an increased interaction of PDHA1 with SIRT3 in the infection scenario in comparison to the uninfected control, which gets eventually abolished under the knockdown condition (*Figure 7H and I*) and under the chemical inhibitor treatment of SIRT3 (*Figure 7J, K and L*) suggesting the role of the SIRT3 in mediating the interaction with PDHA1. Alongside the decreased interaction of PDHA1 with SIRT3, increased acetylation of PDHA1 was detected upon SIRT3 inhibitor treatment in infected macrophages (*Figure 7K and L*).

## SIRT1 or SIRT3 inhibition enhances bacterial burden in mice in vivo

Here, 6-8 weeks old adult male C57BL/6 mice were treated with SIRT1 inhibitor EX-527, SIRT3 inhibitor 3TYP and SIRT1 activator SRT1720 at a dose of 1mg/kg each via intraperitoneal injection (every alternate Day) (*Figure 8A*). Following the inhibitor treatment, the mice were orally gavaged with $10^7$ CFU of *S.* Typhimurium 14028S for organ burden evaluation or with $10^8$ CFU of wildtype *S.* Typhimurium for survival studies. On day 5th post-infection, mice were sacrificed, and the liver, spleen and Mesenteric Lymph Node (MLN) were harvested for enumeration of the organ burden. The SIRT1 inhibitor, EX-527 and SIRT3 inhibitor, 3TYP-treated mice cohorts exhibited increased organ loads in liver, spleen and MLN in comparison to the vehicle control. On the contrary, the SRT1720 treated mice group showed organ burden comparable to that of the vehicle control (*Figure 8B*). Moreover, the SIRT1 and the SIRT3 inhibitor-treated mice cohorts died earlier than the vehicle-treated control mice group or the SIRT1 activator-treated group (*Figure 8C*). Further, the SIRT1 and the SIRT3 inhibitor-treated mice cohorts showed increased splenic length in comparison to the vehicle-treated mice group and the SIRT1 activator-treated mice cohort (*Figure 8D*). The increased organ burden in the EX-527 or 3TYP treated group might be due to increased bacterial dissemination in blood. To assess bacterial dissemination, blood was collected from infected mice post-inhibitor treatment at specific days post-infection retro-orbitally and plated onto *Salmonella Shigella* (SS) agar plates for bacterial enumeration. Indeed, increased bacterial dissemination was observed in the blood of mice treated with SIRT1 inhibitor, EX-527 or SIRT3 inhibitor, 3TYP at day 1-, 2-, 3- ,4- post-infection in comparison to the vehicle-treated mice (*Figure 8E*). Further, we wanted to examine whether the increased bacterial dissemination was due to increased ROS production or due to the presence of elevated inflammatory cytokine levels like IL-6 and IL-1β. In the wildtype C57BL/6 mice treated with SIRT3 inhibitor 3TYP showed heightened bacterial burden in blood at 5th day post-infection in comparison to the vehicle control. Nevertheless, the *Cybb-/-* mice group lacking the catalytic subunit of NADPH oxidase did not depict significant variation in the bacterial load among the different mice treatment cohorts (*Figure 8F*). Further, the EX-527 (SIRT1 inhibitor) and the 3TYP (SIRT3 inhibitor) treated mice possessed elevated levels of serum IL-6 (*Figure 8G*), IL-1β (*Figure 3—figure supplement 1F*) and showed increased intracellular ROS burden in infected liver tissues in comparison to the vehicle-treated control and the SRT1720 (SIRT1 activator) treated mice group (*Figure 8H*). The increased mouse serum IL-6 and IL-1β production was in a similar line with the increased IL-6 or IL-1β cytokine generation in EX-527 or 3TYP treated peritoneal macrophages under the infection scenario (*Figure 3—figure supplement 1D-E*). Moreover, estimation of IL-1β within the infected intestinal ileal sections of the mice revealed increased pro-inflammatory IL-1β generation in the SIRT1 and SIRT3 inhibitor-treated mice groups in comparison to the untreated or the SIRT1-activator treated mice cohorts (*Figure 3—figure supplement 1G*). However, contrary to the in vitro studies wherein *Sirt1* or *Sirt3* knockdown or inhibition resulted in attenuated intracellular

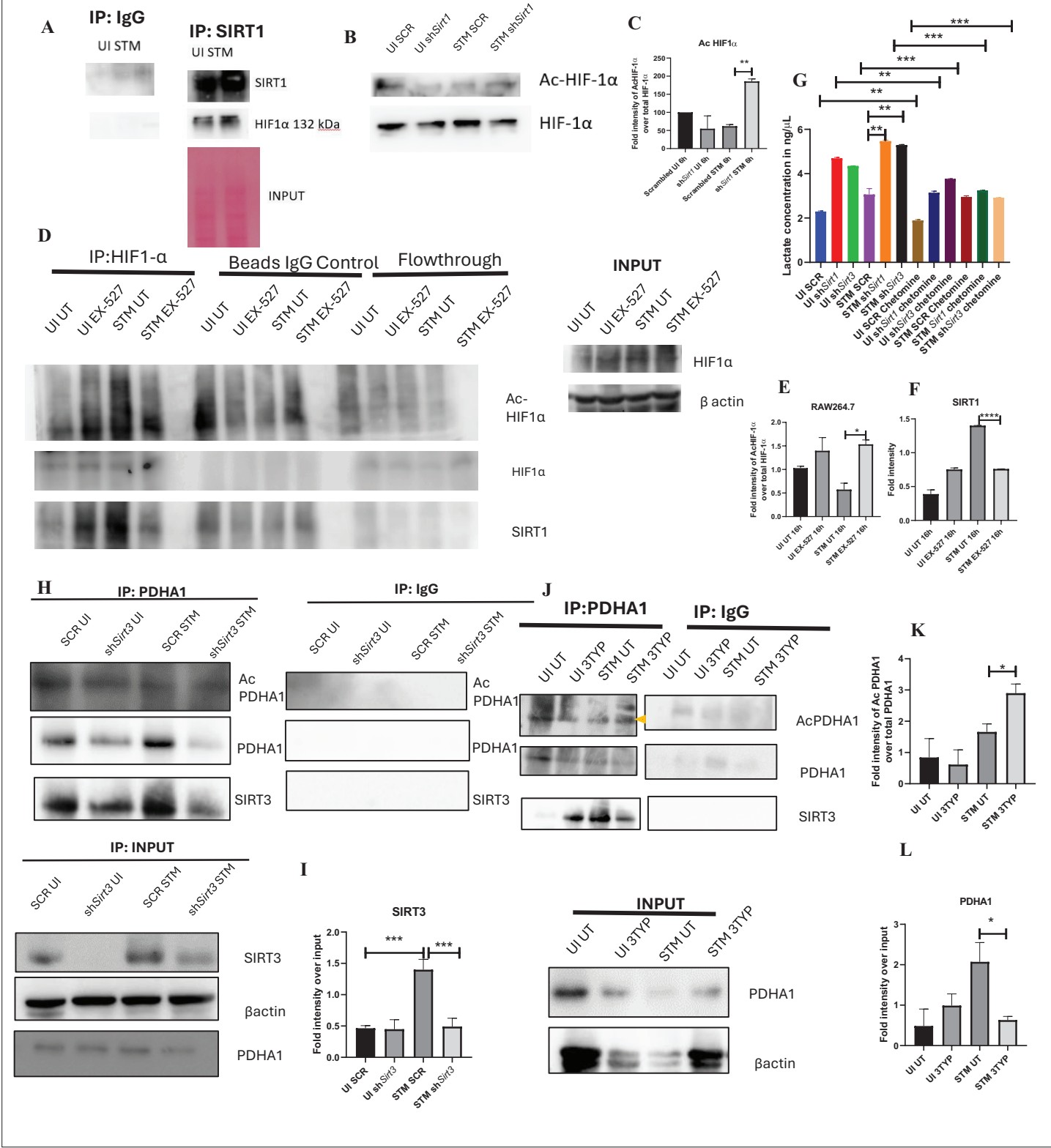

**Figure 7.** SIRT1 inhibition triggers hyperacetylation of glycolytic master regulator HIF-1α within *S.* Typhimurium (STM)-infected macrophages while SIRT3 skews metabolism of *S.* Typhimurium-infected macrophages via interaction with PDHA1. (**A**) An immunoblot depicting HIF-1α interaction with SIRT1 post immunoprecipitation of SIRT1 in uninfected (UI) or STM-infected RAW 264.7 macrophages at 16 hr post-infection (derived from the same SIRT1 IP blot as in *Figure 4A*). (**B**) Immunoblotting of HIF-1α in SIRT1 knockdown UI or STM-infected RAW 264.7 cells at 16 hr post-infection to assess the acetylation status of HIF-1α. (**C**) Densitometric plot depicting the band intensities of acetylated HIF-1α over total HIF-1α in blot **B**. Data is

*Figure 7 continued on next page*

*Figure 7 continued*

representative of N = 3. (**D**) An immunoblot depicting HIF-1α interaction with SIRT1 as well as the HIF-1α acetylation status post immunoprecipitation of HIF-1α [IP:HIF-1α or with control IgG (IP:IgG)] in UI or STM-infected RAW 264.7 macrophages upon SIRT1 inhibitor (EX-527, 1 μM) treatment at 16 hr post-infection. UT, untreated. (**E**) Densitometric plot depicting the band intensities of acetylated HIF-1α over total HIF-1α in blot **D**. Data is representative of N = 3. (**F**) Densitometric plot depicting the band intensities of SIRT1 in blot **D**. (**G**) Lactate estimation assay of STM-infected RAW 264.7 macrophages upon *Sirt1* and *Sirt3* knockdown condition in the presence of HIF-1α inhibitor, chetomine (50 nM) at 16 hr post-infection. Data is representative of N = 3, n = 3. Unpaired two-tailed Student's *t*-test was performed to obtain the p-values (****p<0.0001, ***p<0.001, **p<0.01, *p<0.05). (**H**) An immunoblot depicting PDHA1 interaction with SIRT1 or SIRT3 post immunoprecipitation of PDHA1 in UI or STM-infected RAW 264.7 macrophages in SIRT3 knockdown condition at 16 hr post-infection. (**I**) Densitometric plot depicting the band intensities of SIRT3 interaction over total input in blot. Data is representative of N = 3. (**J**) An immunoblot depicting PDHA1 interaction with SIRT3 post immunoprecipitation of PDHA1 in UI or STM-infected RAW 264.7 macrophages at 16 hr post-infection under SIRT3 inhibitor (3-TYP, 1 μM) treatment at 16 hr post-infection. UT, untreated. (**K**) Densitometric plot depicting the band intensities of acetylated PDHA1 over total PDHA1 in blot **I**. Data is representative of N = 3. (**L**) Densitometric plot depicting the band intensities of SIRT3 interaction over total input in blot **I**. Data is representative of N = 3.

The online version of this article includes the following source data for figure 7:

**Source data 1.** Original files for western blot analysis displayed in *Figure 7A*.

**Source data 2.** Original files for western blot analysis displayed in *Figure 7B*.

**Source data 3.** Original files for western blot analysis displayed in *Figure 7D*.

**Source data 4.** Original files for western blot analysis displayed in *Figure 7H*.

**Source data 5.** Original files for western blot analysis displayed in *Figure 7J*.

**Source data 6.** File containing original uncropped western blots for *Figure 7A*, indicating relevant bands and treatments.

**Source data 7.** File containing original uncropped western blots for *Figure 7B*, indicating relevant bands and treatments.

**Source data 8.** File containing original uncropped western blots for *Figure 7D*, indicating relevant bands and treatments.

**Source data 9.** File containing original uncropped western blots for *Figure 7H*, indicating relevant bands and treatments.

**Source data 10.** File containing original uncropped western blots for *Figure 7J*, indicating relevant bands and treatments.

proliferation, here in in vivo mouse model of infection, we observed increased bacterial organ loads owing to increased bacterial dissemination. To delineate this observation further, we evaluated the bacterial load within splenocytes isolated from control or inhibitor-treated C57BL/6 mice infected with GFP expressing *S.* Typhimurium at 5th day post-infection via flow cytometry. We observed heightened bacterial load in the EX-527 or the 3TYP treated mice cohorts (*Figure 9A-B*). However, when we enumerated the bacterial count within the F4/80+ macrophage population of the infected spleno-cytes, we noticed decreased bacterial loads in the EX-527 or 3TYP -treated mice group in comparison to the vehicle-treated control group or the SRT-1720 activator-treated group (*Figure 9C-D*). Further, we evaluated additional splenic populations including CD45+, Ly6C+, and CD11c+ populations. Our results show that the CD45+ splenic population depicts increased bacterial loads like that of the total splenic population within the SIRT1 or SIRT3 inhibitor-treated cohorts. However, CD45+ monocytes and Ly6C positive splenic population exhibit compromised burden within the SIRT1 and SIRT3 inhibitor-treated cohorts. Moreover, CD11c+ population, CD45+ granulocytes, or lymphocytes show comparable organ loads to that of the vehicle control or SIRT1 activator-treated mice group (*Figure 9E-K*, *Figure 9—figure supplement 1*, *Figure 9—figure supplement 2*, *Figure 9—figure supplement 3*). Overall, our data suggest heterogeneous bacterial burden in diverse splenic populations. This opposing phenotype could be attributed to the increased IL-6 and IL-1β cytokine storm and elevated ROS production upon the SIRT1 or SIRT3 inhibitor treatment which in turn resulted in bacterial dissemination in vivo and concomitantly restricted the in vitro intracellular proliferation within macrophages. To validate this observation, we estimated the ROS levels within the liver and spleen tissues harvested from *S.* Typhimurium infected C57BL/6 mice, treated with specific catalytic inhibitor, activator or vehicle via DCFDA staining using flow cytometry at 5th day post-infection. We detected escalated levels of ROS within both the infected liver and spleen tissues of the EX-527 or 3TYP-treated mice groups in comparison to the vehicle-treated or the SRT1720 treated mice cohorts (*Figure 8H*, *Figure 8—figure supplement 1*). Haematoxylin and eosin staining of the liver sections (harvested at 5th day post-infection) revealed increased inflammation with multiple areas of severe acute hepatic necrosis with complete loss of hepatic architecture in the EX-527 and 3-TYP treated liver samples in comparison to the vehicle-treated control and SRT-1720 treated liver samples (*Figure 10A-B*). In line with the inhibitor-treated studies, the increased organ loads, and systemic dissemination

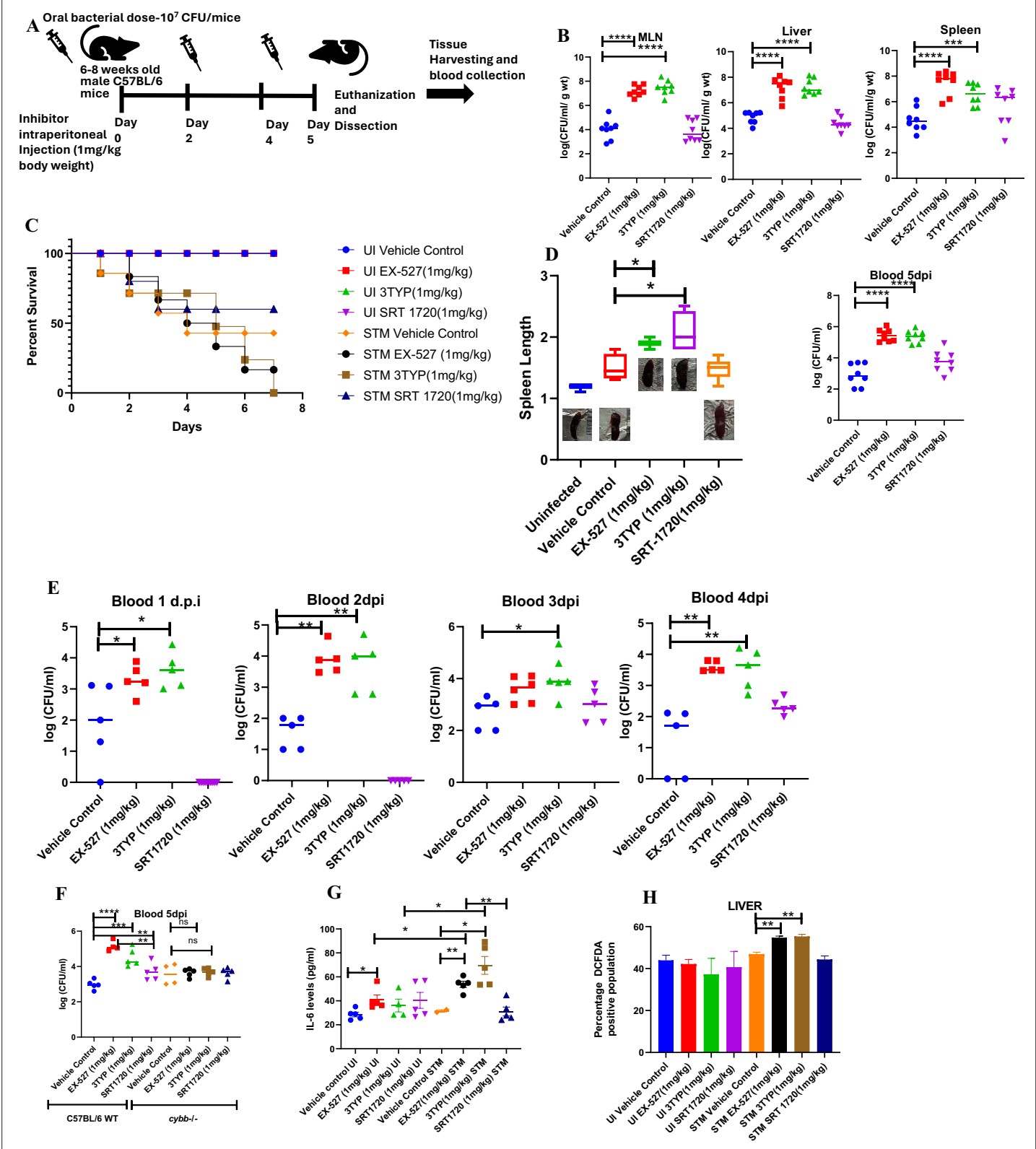

**Figure 8.** Effect of SIRT1 or SIRT3 inhibition on *S.* Typhimurium (STM)-infected C57BL/6 mice. (**A**) The schematic representation of the experimental strategy for studying the effect of SIRT1 and SIRT3 on the in vivo pathogenesis of STM (WT). (**B**) In vivo organ burden of STM upon SIRT1 or SIRT3 inhibition in C57BL/6 mice on fifth day post- infection under specified dosage of inhibitor treatment. Data is representative of N = 4, n = 8. Mann Whitney test was performed to obtain the p- values (****p<0.0001, ***p<0.001, **p<0.01, *p<0.05). (**C**) Percent survival of STM- infected C57BL/6

*Figure 8 continued on next page*

*Figure 8 continued*

mice upon SIRT1 or SIRT3 inhibitor treatment at a specific dose of inhibitor treatment. Data is representative of N = 4, n = 5. (**D**) Representation of splenic length of STM- infected spleen tissue harvested from C57BL/6 mice (males) on fifth day post- infection upon SIRT1 or SIRT3 inhibition at 1 mg/kg dosage. Data is representative of N = 4, n = 8 (**p<0.01, *p<0.05). (**E**) Bacterial load in blood at different days post- infection upon SIRT1 or SIRT3 inhibition at 1 mg/kg dosage in C57BL/6 mice (males). Data is representative of N = 3, n > 5 (**p<0.01, *p<0.05) Mann–Whitney test was performed to obtain the p- values (**p<0.01, *p<0.05). (**F**) Bacterial load in blood on fifth day post- infection upon SIRT1 or SIRT3 inhibition at 1 mg/kg dosage in C57BL/6 WT mice (males) and *cybb-/-* (males) mice. Data is representative of N = 3, n > 5. Mann–Whitney test was performed to obtain the p- values (**p<0.01, *p<0.05). (**G**) Serum IL- 6 levels of STM- infected C57BL/6 WT mice (males) mice treated with SIRT1(EX- 527) or SIRT3 (3TYP) inhibitors or SRT1720 (SIRT1 activator) at 1 mg/kg dosage on fifth day post- infection. Data is representative of N = 3, n > 5. Unpaired two- tailed Student's t- test was performed to obtain the p- values (**p<0.01, *p<0.05). (**H**) Quantitative analysis of percentage population of cells within liver showing DCFDA- positive staining shown in*Figure 8—figure supplement 1A*.Data is representative of N = 3, n > 5. Unpaired two- tailed Student's t- test was performed to obtain the p- values (** p<0.01).

The online version of this article includes the following figure supplement(s) for figure 8:

**Figure supplement 1.** SIRT1 and SIRT3 inhibition led to enhanced ROS production within *Salmonella*-infected mice liver and spleen tissues.

driven heightened susceptibility of mice toward *S.* Typhimurium infection were replicated in vivo *Sirt1* and *Sirt3* adeno-associated virus serotype 6 (AAV6) mediated knockdown mice model which showed elevated IL-6 production in comparison to the scrambled control treated mice cohort (*Figure 10C-F*, *Figure 10—figure supplement 1*). Simultaneously, the haematoxylin-eosin-stained sections of the liver tissues harvested from the sh*Sirt1* or sh*Sirt3* mice cohorts depicted increased pathological scoring with multiple necrotic areas and severely damaged liver tissue architecture in comparison to the scrambled mice control (*Figure 10F*). Altogether, our results implicate the role of SIRT1 and SIRT3 in controlling *S.* Typhimurium infection in vivo.

## Discussion

Several studies have confirmed the propensity of *Salmonella* to skew the polarization state of the infected macrophages toward an immunosuppressive anti-inflammatory state (*Panagi et al., 2020*; *Stapels et al., 2018*; *Pham et al., 2020*). We have validated such findings and further elaborated it by depicting the role of SIRT1 and SIRT3 in the modulation of host immune responses as well as host–bacterial metabolism. *S.* Typhimurium infection modulates the expression profile of both SIRT1 and SIRT3 in the infected macrophages at both mRNA and protein level via its SPI-2 effector. Downregulation of *Sirt1* and *Sirt3* through shRNA-mediated knockdown resulted in heightened pro-inflammatory immune responses with increased production of IL-6, IL-1β cytokines and decreased surface expression of anti-inflammatory CD206. SIRT1 and SIRT3 downregulation also resulted in increased intracellular ROS production in the infected macrophages. *Sirt1* and *Sirt3* knockdown macrophages not only show altered host immune status but also depict shift in the metabolic state with increased glycolytic shift. This altered host metabolism upon *Sirt1* and *Sirt3* knockdown condition influences the outcome of infection by regulating the intracellular bacterial metabolism, which shows reduced bacterial glycolysis and increased FAO. All these outcomes account for attenuated intracellular bacterial proliferation in the *Sirt1* and *Sirt3* knockdown macrophages. However, in murine model of infection, SIRT1 or SIRT3 inhibitor treatment led to increased organ burden and triggered bacterial dissemination (*Figure 10—figure supplement 2*). Overall, our study highlights the crucial role of SIRT1 and SIRT3 in governing the host immune-metabolic shift during *Salmonella* infection, which in turn is vital for maintaining *Salmonella* metabolism.

Previous reports have elucidated the role of SIRT1 and SIRT2 pertaining to *Salmonella* infection. Ganesan et al. have depicted the role of SIRT1 in autophagy in *Salmonella* infection scenario (*Ganesan et al., 2017*). Gogoi et al. have demonstrated SIRT2-mediated modulation of immune responses in dendritic cells (*Gogoi et al., 2018*). To date, there is no report highlighting the role of SIRT3 governing the *Salmonella* pathogenesis. The function of SIRT3 in infection scenario has been explored quite recently. In *Mycobacterium tuberculosis* infection condition, SIRT3 control mitochondrial function and autophagy (*Kim et al., 2019*). SIRT3 downregulation in *M. tuberculosis*-infected macrophages

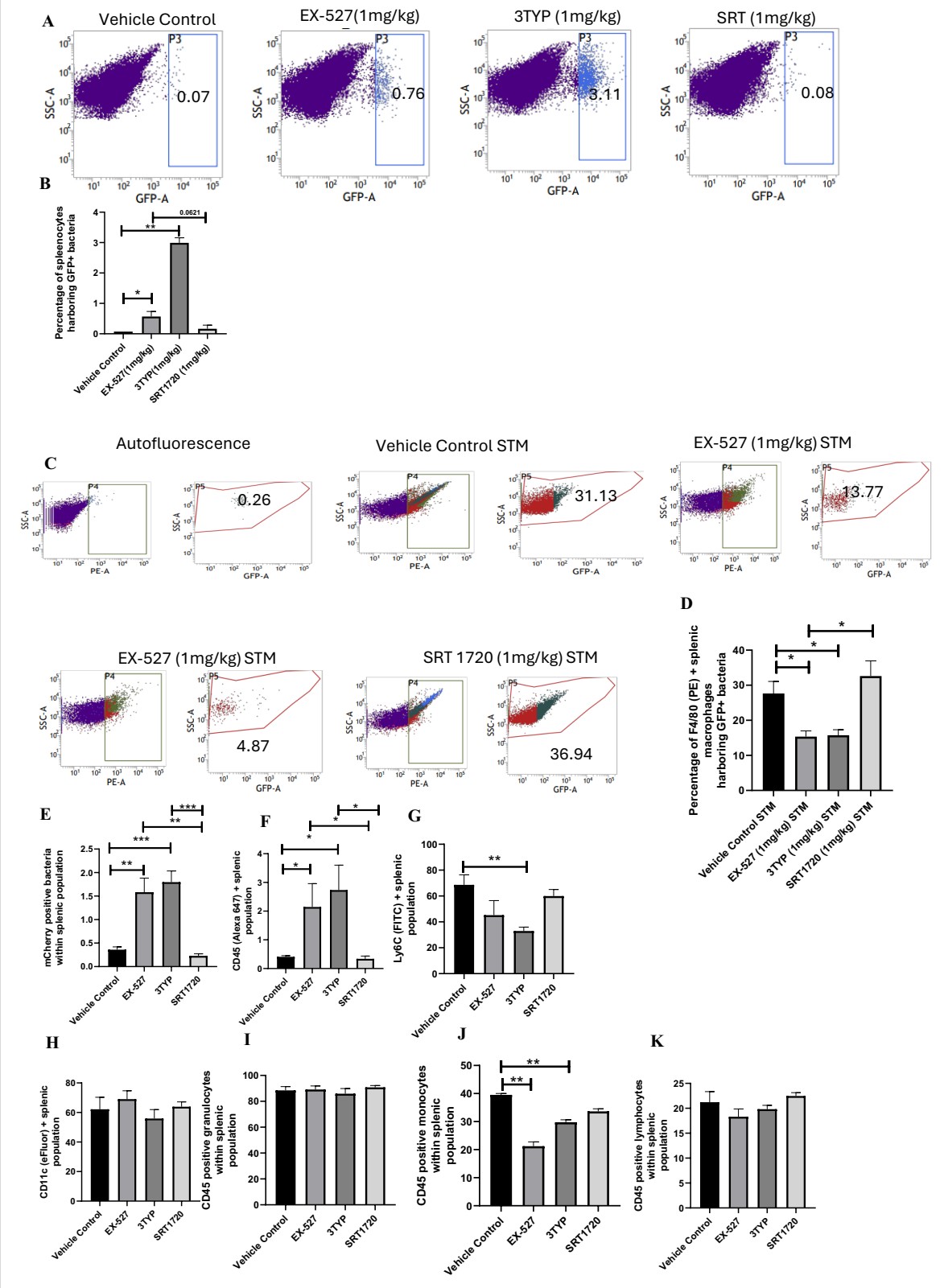

**Figure 9.** Effect of SIRT1 and SIRT3 inhibition on bacterial loads in diverse populations of *S.* Typhimurium-infected splenocytes. (**A**) Enumeration of GFP-positive bacterial cells through flow cytometry in splenic tissues homogenate harvested from adult male 6- to 8-week-old C57BL/6 mice (subjected to different chemical treatment – vehicle treated or SIRT1 [EX-527] or SIRT3 [3-TYP] inhibitor or SIRT1 activator SRT1720 treated at a dose of 1 mg/kg) on fifth day post STM (expressing GFP) infection (10⁷ CFU orally). Data is representative of N = 3, n > 5. Unpaired two-tailed Student's *t*-test was performed

*Figure 9 continued*

to obtain the p-values (\*\*p<0.01). (**B**) Quantitative analysis of the percentage population of splenic cells harbouring GFP+ bacterial cells shown in (**A**). Unpaired two-tailed Student's *t*-test was performed (\*\*p<0.01, \*p<0.05). (**C**) Enumeration of GFP-positive bacterial cells through flow cytometry within F4/80-positive splenic macrophages present within splenic tissues homogenate harvested from adult male 6- to 8-week-old C57BL/6 mice (subjected to different chemical treatment – vehicle treated or SIRT1 [EX-527] or SIRT3 [3-TYP] inhibitor or SIRT1 activator SRT1720 treated at a dose of 1 mg/kg) on fifth day post STM infection. Data is representative of N = 3, n > 5. (**D**) Quantitative analysis of percentage population of F4/80-positive macrophage cells harbouring GFP+ bacteria shown in (**C**). Unpaired two-tailed Student's *t*-test was performed to obtain the p-values. Data is representative of N= 3, n > 5 (\*p<0.05). (**E-K**) Quantitative analysis of different mCherry-*S*. Typhimurium-infected splenic populations harbouring mCherry+ bacteria via flow cytometry depicted in *Figure 9—figure supplement 1*.

The online version of this article includes the following figure supplement(s) for figure 9:

**Figure supplement 1.** Flow cytometric analysis of different splenic populations within mCherry-labelled *S*. Typhimurium-infected mice cohorts upon SIRT1 or SIRT3 inhibitor treatment.

**Figure supplement 2.** Flow cytometric analysis of different splenic populations within mCherry- labelled *S*. Typhimurium- infected mice cohorts upon SIRT1 or SIRT3 inhibitor treatment.

**Figure supplement 3.** Flow cytometric analysis of different splenic populations within mCherry- labelled *S*. Typhimurium- infected mice cohorts upon SIRT1 or SIRT3 inhibitor treatment.

is associated with dysregulated mitochondrial metabolism and increased cell death (*Smulan et al., 2021*). In this study, we have explored the role of SIRT1 and SIRT3 in mediating host immune-metabolic switch in *S*. Typhimurium-infected macrophages, which further govern intracellular bacterial metabolism and pathogenesis.

Our findings suggest the role of SIRT1 and SIRT3 in mediating polarization of the *Salmonella*-infected macrophages toward an anti-inflammatory state. Upon knockdown of *Sirt1* and *Sirt3* in the infected macrophages, we detect robust pro-inflammatory response and oxidative burst. This is in line with the findings by Elsela et al., wherein SIRT1 knockout respiratory syncytial virus (RSV)-infected Bone marrow derived dendritic cells, BMDCs showed significant increase in *Il1b*, *Il6,* and *Il23* expression and ROS generation in comparison to the wildtype RSV-infected BMDCs (*Elesela et al., 2020*). Also, Kim et al. showed presence of aggravated inflammatory responses in *M. tuberculosis*-infected *Sirt3*-/- Bone marrow derived macrophages, BMDMs (*Kim et al., 2019*). This heightened pro-inflammatory cytokine and oxidative burst restrict the intracellular survival of the pathogen as detected by the lower intracellular bacterial burden in the *Sirt1* and *Sirt3* knockdown murine macrophages. *Salmonella* showed enhanced proliferation in the M2 macrophages owing to the reduced arsenals in terms of pro-inflammatory cytokines and ROS production. Moreover, the M2 macrophages are fuelled by increased FAO and reduced glycolysis (*Eisele et al., 2013*). This might facilitate enhanced bacterial proliferation as the host unutilized intracellular glucose can be readily up taken by the pathogen and used to support its own glycolysis. Similarly, M1 or pro-inflammatory macrophages resort to glycolysis to meet their energy demands (*Cramer et al., 2003*), thereby limiting the glucose availability for the intracellular pathogen (*Cramer et al., 2003*; *Merrill et al., 1997*). In such conditions, bacteria show enhanced fatty acid metabolism to sustain their energy demand (*Reens et al., 2019*). In our study, we found that wildtype *S*. Typhimurium infection drives host metabolism toward increased FAO via its SPI-2 effector protein with concomitant increase in the bacterial glycolysis. SIRT1 and SIRT3 inhibition abrogates the metabolic switch and triggers increase in host glycolysis, which in turn skew the bacterial metabolism from increased glycolysis to enhanced FAO and reduced glycolysis. Together, these findings implicate the role of SIRT1 and SIRT3 in reprogramming the host metabolism, which in turn affect the intracellular nutrient niche of the pathogen, thereby influencing intracellular *Salmonella* proliferation. However, our in vivo findings in the murine model of infection show increased bacterial burden upon SIRT1 or SIRT3 inhibition. This increased burden could be attributed to increased dissemination from the macrophages into the bloodstream owing to the increased level of serum IL-6 levels. This is in concert with previous findings in *Klebsiella pneumoniae* infection in mice wherein increased inflammatory response upon HIF-1α activation induces bacterial dissemination (*Holden et al., 2016*). Further correlation analysis of immune responses to *Salmonella* infection revealed that increased innate immune 'cassette' opposes the adaptive immune arm, leading

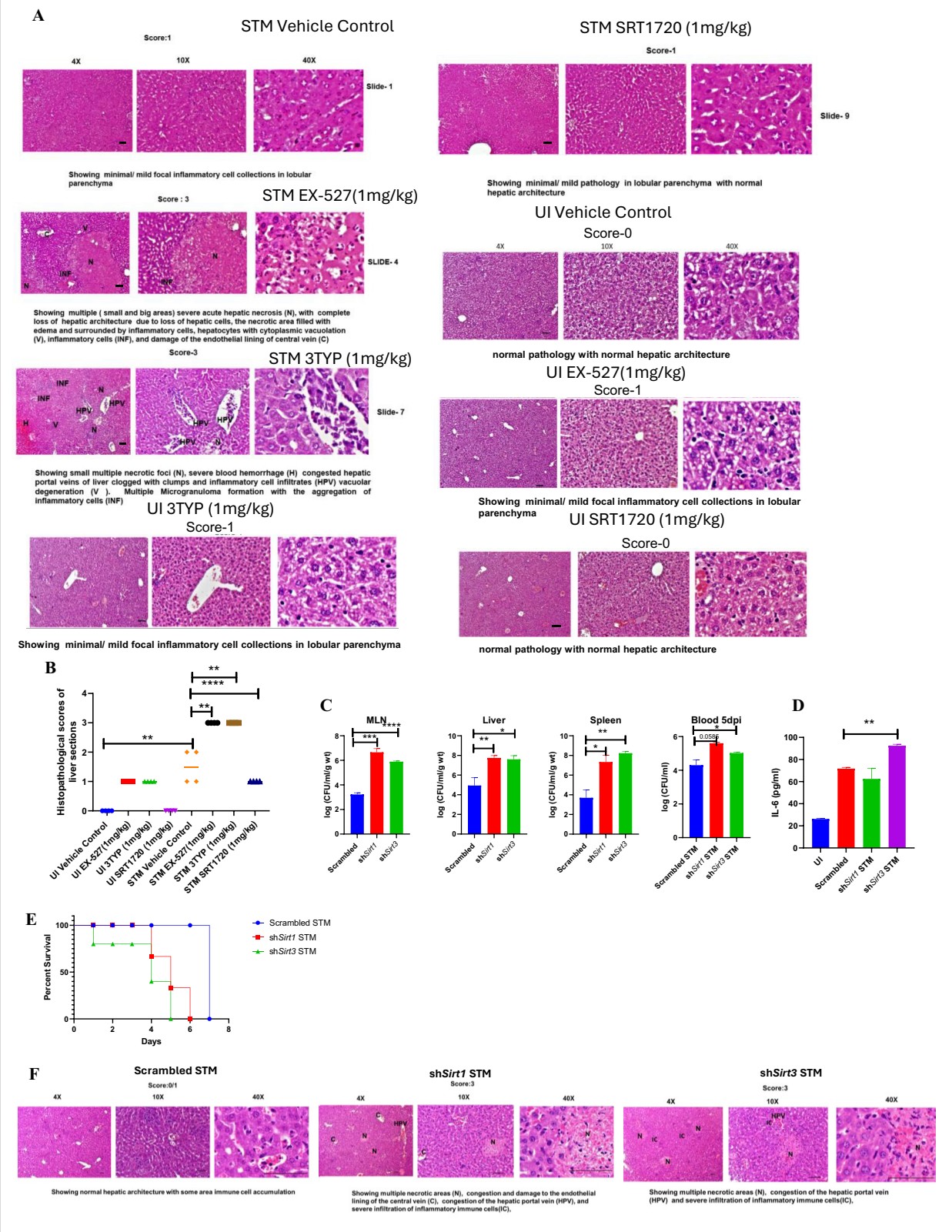

**Figure 10.** In vivo role of SIRT1 and SIRT3 inhibition and knockdown during *S*. Typhimurium infection. (**A**) Representative image of haematoxylin-eosin-stained liver sections to assess the liver tissue architecture upon *Salmonella* infection on fifth days post-infection in different mice cohorts (UI, uninfected; STM, *S*. Typhimurium-infected, EX-527-SIRT1 inhibitor, 3TYP-SIRT3 inhibitor, SRT1720- SIRT1 activator, vehicle control, PBS containing 0.1% DMSO). Scale bar, 50 µm. **Scoring system:** according to pathological changes, the tissue sections are scored as 0 for normal pathology, 1 for mild/minor

*Figure 10 continued on next page*

*Figure 10 continued*

pathology, 2 for moderate pathology, and 3 for severe pathological changes. (**B**) Graph representing the histopathological scoring of the liver sections depicted in (**A**). Unpaired two-tailed Student's *t*-test was performed to obtain the p-values (*p<0.05, ****p<0.0001, ***p<0.001, **p<0.01). (**C**) In vivo organ burden of STM upon *Sirt1* or *Sirt3* adenovirus-mediated in vivo knockdown in C57BL/6 mice on fifth day post-infection. Data is representative of N = 3, n > 3. Mann–Whitney test was performed to obtain the p-values (****p<0.0001, *** p<0.001, ** p<0.01, * p<0.05). (**D**) Serum IL-6 levels of STM-infected C57BL/6 WT mice (males) upon in vivo adenovirus-mediated *Sirt1* or *Sirt3* knockdown. Data is representative of N = 3, n > 3. Unpaired two-tailed Student's *t*-test was performed to obtain the p-values (**p<0.01, *p<0.05). (**E**) Percent survival of STM -infected C57BL/6 mice upon in vivo adenovirus-mediated *Sirt1* or *Sirt3* knockdown. Data is representative of N = 3, n > 3. (**F**) Representative image of haematoxylin-eosin-stained liver sections upon *Salmonella* infection at 5th days post-infection in different mice cohorts. (Scrambled STM, sh*Sirt1* STM, sh*Sirt3* STM). Scale bar-50µm. Scoring system:- according to pathological changes the tissue sections are scored as 0 for normal pathology, 1 for mild/ minor pathology, 2 for moderate pathology, and 3 for severe pathological changes.

The online version of this article includes the following source data and figure supplement(s) for figure 10:

**Figure supplement 1.** In vivo knockdown validation by western blotting and qPCR in liver.

**Figure supplement 2.** Schematic diagram – in murine macrophages, *Salmonella* infection drives an immuno-metabolic shift toward an immunosuppressive state with an increase in host fatty acid oxidation rate by modulating SIRT1 and SIRT3.

**Figure supplement 1—source data 1.** Original files for western blot analysis displayed in *Figure 10—figure supplement 1A*.

**Figure supplement 1—source data 2.** File containing original uncropped western blots for *Figure 10—figure supplement 1A*, indicating relevant bands and treatments.

to increased bacterial load (*Hotson et al., 2016*). Moreover, previous literature studies suggested that a low dose of Sirt1 activator such as resveratrol treatment in rats for 25-day treatment under 5% dextran sulphate sodium-induced colitis condition led to alterations in gut microbiota profile with increased lactobacilli and bifidobacteria alongside a reduced abundance of enterobacteria (*Lakhan and Kirchgessner, 2011*; *Larrosa et al., 2009*). This study correlates with our study wherein we have detected enhanced *Salmonella* (belonging to Enterobacteriaceae family) loads under both SIRT1/3 in vivo knockdown or inhibitor-treated condition in C57BL/6 mice alongside reduced burden under SIRT1 activator, SRT1720 treatment.

Future studies might explore the host and bacterial interacting partners of SIRT1 and SIRT3 through mass spectrometry analyses in *Salmonella*-infected macrophages, which might hint at the underlying mechanism of their action and regulation. Together, this study highlights the complex and multifaceted nature of host–pathogen interactions, and the need for further research to fully understand the role of SIRT1 and SIRT3 in the context of *Salmonella* infection.

# Materials and methods

## Key resources table

| Reagent type (species) or resource | Designation | Source or reference | Identifiers | Additional information |
|---|---|---|---|---|
| Strain, strain background | *Salmonella enterica* serovar Typhimurium ATCC strain 14028S (STM) | ATCC strain 14028S – gfted by Prof. M. Hensel | ATCC strain 14028S | |
| Recombinant DNA reagent | pFPV:GFP-Amp$^R$ | Laboratory stock | | |
| Recombinant DNA reagent | pFPV:mCherry-Amp$^R$ | Laboratory stock | | |
| Recombinant DNA reagent | pLKO.2: shRNA-Amp$^R$ | Laboratory stock – gifted by Prof. Subba Rao | | |
| Recombinant DNA reagent | STM Δ *invC*-Kan$^R$ | Laboratory stock | | |
| Recombinant DNA reagent | STM Δ *ssaV*-Chl$^R$ | Laboratory stock | | |
| Recombinant DNA reagent | STM Δ *steE*-Kan$^R$ | Laboratory stock | | |

*Continued on next page*

*Continued*

| Reagent type (species) or resource | Designation | Source or reference | Identifiers | Additional information |
|---|---|---|---|---|
| Cell line | RAW 264.7; macrophage; mouse (*Mus musculus*) | ATCC | ATCC-TIB-71 | |
| Antibody | SIRT1 polyclonal | Thermo Fisher Scientific | Cat# PA5-85921 | 1:1000 dilution (western), 1:100 (immunoprecipitation) |
| Antibody | SIRT3 monoclonal | Thermo Fisher Scientific | Cat# PA5-13222 | 1:1000 dilution (western), 1:100 (immunoprecipitation) |
| Antibody | CD86 monoclonal | Thermo Fisher Scientific | Cat# 12-086282PE | 1:100 dilution |
| Antibody | CD206 monoclonal | Thermo Fisher Scientific | Cat# 17-2069-42 | 1:100 dilution |
| Antibody | F4/80 monoclonal antibody | BD Horizon | Cat# 565411 | 1:100 dilution |
| Antibody | HIF-1α monoclonal | Santa Cruz | sc-13515 SCBT | 1:1000 dilution (western), 1:100 (immunoprecipitation) |
| Antibody | NF-κβ p65 monoclonal | CST (D14E12) | #8242 | 1:1000 dilution (western), 1:100 (immunoprecipitation) |
| Antibody | Acetylated-lysine antibody polyclonal | CST | Cat# 9441 | 1:1000 dilution |
| Antibody | PDHA1 monoclonal | ABclonal | Cat# A13687 | 1:1000 dilution (western), 1:100 (immunoprecipitation) |
| Antibody | HADHA monoclonal | ABclonal | Cat# A5346 | 1:1000 dilution |
| Antibody | ACOX1 monoclonal | ABclonal | Cat# A8091 | 1:1000 dilution |
| Antibody | PGK monoclonal | ABclonal | Cat# A12686 | 1:1000 dilution |
| Antibody | PFK monoclonal | ABclonal | Cat# A5477 | 1:1000 dilution |
| Antibody | β actin monoclonal | Sigma | A3854 | 1:50,000 dilution |
| Commercial assay or kit | Lactate Assay Kit | Sigma | Cat# MAK064 | |
| Commercial assay or kit | Fatty acid oxidation (FAO) Assay Kit | AssayGenie | Catalogue code BR00001 | |
| Commercial assay or kit | IL-6 | BD Bioscience | Cat# 555240 | |
| Commercial assay or kit | IL-10 | BD Bioscience | Cat# 555252 | |
| Commercial assay or kit | IL-1β | SARD Biosciences | Cat# SB-EKM1085 | |
| Antibody | CD16/32 monoclonal | eBioscience | Cat# 14-0161-82 | 1:100 dilution |
| Antibody | eFluor450-conjugated anti-mouse CD11c monoclonal | eBioscience | Cat# 48-0116-42 | 1:100 dilution |
| Antibody | FITC-conjugated anti-mouse Ly6C antibody monoclonal | BD Pharmingen | Cat# 553104 | 1:100 dilution |
| Antibody | Alexa 647-conjugated anti-mouse CD45 antibody mooclonal | BioLegend | Cat# 103124 | 1:100 dilution |
| Chemical compound, drug | SIRT1 inhibitor EX-527 | Sigma-Aldrich | E7034-5MG | |
| Chemical compound, drug | SIRT3 inhibitor 3TYP | Selleck Chemical | Cat# S8628 | |
| Chemical compound, drug | SIRT1 activator SRT1720 | Sigma-Aldrich | 567860-10MG | |

*Continued on next page*

*Continued*

| Reagent type (species) or resource | Designation | Source or reference | Identifiers | Additional information |
|---|---|---|---|---|
| Chemical compound, drug | N-Acetyl cysteine | Sigma-Aldrich | A9165 | |
| Chemical compound, drug | PEI Max 4000 | Polysciences | Cat# 24765 | |
| Chemical compound | Protease Inhibitor Cocktail | Roche | 04693132001 | |
| Chemical compound, drug | Saponin | Sigma | SAE0073 | |
| Chemical compound, drug | Chetomin | Sigma | C9623 | |
| Chemical compound, drug | SS agar | Himedia | M-108 | |
| Chemical compound, drug | Skimmed milk | Himedia | GRM-1254 | |

## Bacterial strains and culture conditions

*Salmonella enterica* serovar Typhimurium (STM) strain ATCC 14028S or ATCC 14028S constitutively expressing green fluorescent protein (eGFP) or mCherry (RFP) through pFPV plasmid were used. 4% paraformaldehyde-fixed STM (PFA) was used as the killed fixed bacteria control. The abovementioned live bacterial strain was grown overnight in LB broth in 37°C at 160 rpm shaking condition in the presence or absence of appropriate antibiotic after revival of the bacterial strains from glycerol stock (stored at –80°C).

## Cell culture

RAW 264.7 (ATCC) murine macrophages were cultured in Dulbecco's Modified Eagle Medium, DMEM (Lonza) containing 10% Fetal Bovine Serum, FBS (Gibco) at 37°C in a humidified incubator with 5% $CO_2$. Prior to each experiment, cells were seeded into 24-well or 6-well plate as per requirement at a confluency of 60%.

For macrophage polarization experiments, the seeded macrophages were subjected to 100 ng/ml LPS+ 20 ng/ml IFN-γ treatment for M1 polarization and 20 ng/ml IL-4 treatment for M2 polarization for 24 hr. Post-polarization, the cell supernatant was collected for validation of polarization status by ELISA and was further subjected to infection protocol.

Peritoneal macrophages were collected in Phosphate Buffered Saline (PBS) from the peritoneal cavity of 6- to 8-week-old adult male C57BL/6 mice aseptically post thioglycolate treatment using 20G needle and 5 ml syringe. Following centrifugation, cell pellet was resuspended in RPMI-1640 (Lonza) containing 10% heat-inactivated FBS (Gibco), 100 U/ml penicillin, and 100 µg/ml streptomycin and

**Table 1.** List of shRNA used for knockdown.

| Sirtuins | Construct ID | TRC ID | Sequence (5'–3') |
|---|---|---|---|
| *Sirt1* | C4 | TRCN0000218734 | GTACCGGCATGAAGTGCCTCAGATATTACTCGAGTAATATCTGAGGCACTTCATGTTTTTTG |
| *Sirt1* | C12 | TRCN0000018979 | CCGGGCAAAGCCTTTCTGAATCTATCTCGAGATAGATTCAGAAAGGCTTTGCTTTTT |
| *Sirt3* | E8 | TRCN0000038889 | CCGGCCCAACGTCACTCACTACTTTCTCGAGAAAGTAGTGAGTGACGTTGGGTTTTTG |
| *Sirt3* | E12 | TRCN0000038893 | CCGGCCACCTGCACAGTCTGCCAAACTCGAGTTTGGCAGACTGTGCAGGTGGTTTTTG |

seeded into 6-well plate. Then, 6 hr prior to infection, antibiotic-containing media was replaced with Penicillin-Streptomycin free RPMI-1640 (Lonza) containing 10% heat-inactivated FBS (Gibco).

## Transfection

shRNA-mediated knockdown was carried out by PEI-mediated transfection protocol. Plasmid harbouring shRNA in pLKO.2 vector backbone specific to *Sirt1* and *Sirt3* were used for transfection (*Table 1*). Plasmid harbouring scrambled sequence of shRNA, served as a control, was also used for transfection. Plasmid DNA was used at a concentration of 500 ng and 1 µg per well of a 24-well plate and 6-well plate, respectively. Plasmid and PEI were added in 1:2 ratio in serum-free DMEM media and incubated for 20 min at room temperature (RT). Post incubation, the DNA: PEI cocktail was added to the seeded RAW 264.7 macrophages. After 6–8 hr of incubation, serum-free media was replaced with complete media containing 10% FBS. Post 48 hr of transfection, transfected cells were either harvested for knockdown confirmation studies or subjected to infection with STM.

## Infection protocol

Macrophages were infected with stationary-phase bacterial culture with MOI of 10. For synchronization of the infection, tissue culture plates were subjected to centrifugation at $600 \times g$ for 5 min and incubated at 37°C humidified incubator with 5% $CO_2$ for 25 min. Cells were washed with PBS and were treated with DMEM (Sigma) + 10% FBS (Gibco) containing 100 µg/ml gentamicin for 1 hr. Subsequently, the gentamicin concentration was reduced to 25 µg/ml and maintained until the cells were harvested. For the inhibitor treatment studies, along with 25 µg/ml containing complete media 1 µM of SIRT1 (EX-527) inhibitor or SIRT3 (3TYP) or 10 mM of NAC (Sigma) or 50 nM of chetomin (Sigma) were added to the cells.

## Immunofluorescence confocal microscopic studies

At the specified time points post-infection with GFP-tagged STM, cells were fixed with 3.5% paraformaldehyde for 15 min. Primary antibody staining was performed with specific primary antibody in the presence of a permeabilizing agent, 0.01% saponin (Sigma) dissolved in 2.5% Bovine serum albumin, BSA containing PBS at 4°C for overnight or for 6 hr at RT. Following this, cells were washed with PBS stained with appropriate secondary antibody tagged with fluorochrome for 1 hr at RT. This was followed by DAPI staining and mounting of the coverslip onto a clean glass slide using the mounting media containing the anti-fade agent. The coverslip sides were sealed with a transparent nail paint. All immunofluorescence images were obtained using Zeiss LSM 710 or Zeiss LSM 880 and were analyzed using ZEN black 2012 software.

## Quantitative real-time PCR

Total RNA was isolated at specific time points post-infection by using TRIzol (Takara) as per the manufacturer's protocol. Quantification of RNA was performed in NanoDrop (Thermo Fisher Scientific). Quality of isolated RNA was detected by performing 2% agarose gel electrophoresis. 2 µg of RNA was subjected to DNaseI (Thermo Fisher Scientific) treatment at 37°C for 1 hr followed by addition of 0.5 M EDTA (final concentration 5 mM) and heat inactivation at 65°C for 10 min. The mRNA was reverse transcribed to cDNA using oligo (dT)$_{18}$ primer, buffer, dNTPs, and reverse transcriptase (Takara) as per the manufacturer's protocol. The expression profile of target gene was evaluated using specific primers (*Table 2*) by using SYBR green RT-PCR master mix (Takara) in Bio-Rad real-time PCR instrument. *Actb* was used as an internal control for mammalian genes and for bacterial genes 16S rRNA was used. All the reaction was setup in 384-well plate with two replicates for each sample.

## Intracellular proliferation or gentamicin protection assay

Following infection of the transfected cells with STM at an MOI of 10, cells were treated with DMEM (Sigma) + 10% FBS (Gibco) containing 100 µg/ml gentamicin for 1 hr. Subsequently, the gentamicin concentration was reduced to 25 µg/ml and maintained until the specified time point. Post 2 hr and 16 hr post-infection, cells were lysed in 0.1% Triton-X-100. Lysed cells were serially diluted and plated on SS agar to obtain colony-forming units (CFUs). Fold proliferation was calculated as CFU at 16 hr divided by CFU at 2 hr.

Fold Proliferation = [CFU at 16h/CFU at 2h]

## Western blotting

Post appropriate time points of infection, the cells were washed in PBS and subsequently harvested in PBS. The cell pellets were obtained after centrifugation at 300 × *g* for 7 min at 4°C. Cells were lysed in 1× RIPA (10×–0.5M NaCl, 0.5 M EDTA pH 8.0, 1 M Tris, NP-40, 10% sodium deoxycholate, 10% SDS) buffer containing 10% protease inhibitor cocktail (Roche) for 30 min on ice. Total protein was estimated using Bradford (Bio-Rad) method of protein estimation. Protein samples were subjected to 12% SDS polyacrylamide gel electrophoresis and then were transferred onto 0.45 μm PVDF membrane (18 V, 2 hr). The membrane was blocked using 5% skim milk in TBST (Tris buffered saline containing 0.1% Tween-20) for 1 hr at RT and subsequently probed with appropriate primary

**Table 2.** List of primers.

| Primer name | Sequence (5'–3') |
| --- | --- |
| *Sirt1* FP | ACAAAGTTGACTGTGAAGCTGTAC |
| *Sirt1* RP | GTTCATCAGCTGGGCACCTA |
| *Sirt3* FP | GCTGCTTCTGCGGCTCTATAC |
| *Sirt3* RP | GAAGGACCTTCGACAGACCGT |
| *Ppard* FP | CACAACGCTATCCGCTTTGG |
| *Ppard* RP | ATGCTCCGGGCCTTCTTTTT |
| *Actb* FP | CAGCAAGCAGGAGTACGATG |
| *Actb* RP | GCAGCTCAGTAACAGTCCG |
| *Acadl* FP | CTTGGGAAGAGCAAGCGTACT |
| *Acadl* RP | CTGTTCTTTTGTGCCGTAATTCG |
| *Hadha* FP | AGCAACACGAATATCACAGGAAG |
| *Hadha* RP | AGGCACACCCACCATTTTGG |
| *Acox1* FP | TCGAAGCCAGCGTTACGAG |
| *Acox1* RP | ATCTCCGTCTGGGCGTAGG |
| *Pdha1* FP | TGTCGGTTCCCAGTCCATCA |
| *Pdha1* RP | CGTTTCCTTTTCACAGCACATGA |
| *Pfkl* FP | GAACTACGCACACTTGACCAT |
| *Pfkl* RP | CTCCAAAACAAAGGTCCTCTGG |
| *ptsG* FP | TATCTGGGCTTCTTTGCGGG |
| *ptsG* RP | ACCAGGCAACGCTCGATAAA |
| *fadA* FP | TCTGGGATTGATGGAGCAGA |
| *fadA* RP | AGACCAATACACATCGTCGC |
| *fadB* FP | GTCCCCGAAGAGCAGTTAAG |
| *fadB* RP | CCAGTTTAGTTTCCGGCAGA |
| *aceA* FP | CGATCTGGTATGGTGCGAAA |
| *aceA* RP | TTCTGCCAGTTGAAGGATGG |
| *aceB* FP | AGCGTTTCAATCAACAGGGT |
| *aceB* RP | CCCGTATTTTTACCTGCCGA |
| *Salmonella* 16S rRNA FP | GTGAGGTAACGGCTCACCAA |
| *Salmonella* 16S rRNA RP | TAACCGCAACACCTTCCTCC |

antibody for overnight at 4°C. Following wash in TBST, blot was probed with specific HRP conjugated secondary antibody for 1 hr at RT. The membrane was developed using ECL (Advansta) and images were captured using ChemiDoc GE healthcare. All densitometric analysis was performed using ImageJ software.

## Immunoprecipitation

For co-immunoprecipitation, cells were washed with PBS and were lysed in native lysis buffer containing 1% Nonidet P-40, 20 mM Tris (pH 8), 2 mM EDTA, 150 mM NaCl, and protease inhibitors mixture (Roche Diagnostics) for 30 min at 4°C. Cell debris was removed by centrifugation at 10,000 rpm for 10 min, and the supernatant was treated with the specific antibody against the protein to be precipitated. Antibody-lysate complexes were immunoprecipitated using Protein A/G- linked magnetic beads (MagGenome) according to the manufacturer's protocol. Beads were extensively washed with washing buffer and denatured at 95°C for 10 min. Denatured precipitates were subjected to SDS-PAGE (12% gel) followed by transfer to 0.45 µ PVDF membrane. The membrane was blocked using 5% skimmed milk in TBST (Tris buffered saline containing 0.1% Tween-20) for 1 hr at RT and eventually probed for the target primary antibodies or anti-acetylated lysine (Ac-K) primary antibody overnight at 4°C. The blot was probed with a specific HRP-conjugated secondary antibody for 1 hr at RT after rigorous washing in TTBS. ECL (Bio-Rad) was used for detection and images were captured using ChemiDoc GE healthcare.

## ELISA

Estimation of cytokines in cell-free supernatant or in mice serum was performed according to the manufacturer's instructions. Briefly, 96-well ELISA plates (BD Bioscience) were coated overnight with capture antibody at 4°C. Following day, plates were washed with 0.1% Tween-20 containing PBS and blocked with 10% FBS for 1 hr. Following blocking, wells were washed and incubated with 100 µl of test samples for 2 hr at RT. Subsequently, plates were washed and incubated with detection antibody and enzyme reagent for 1 hr at RT (BD Bioscience). TMB (Sigma) was used as a substrate and reactions were stopped with 2 N $H_2SO_4$. For the estimation of IL-1β, pre-coated ELISA (SARD Biosciences) plates were used, and ELISA was performed as per the manufacturer's protocol. Absorbance was measured at 450 nm wavelength in Tecan Plate reader, and the concentration of cytokines were interpolated from a standard curve.

## Flow cytometry

After specific time points post-infection, cells were washed and harvested in PBS. Following centrifugation, cell pellet was resuspended in FACS buffer comprised of 1% BSA in PBS. Blocking was performed with Fc blocker (purified Anti-mouse CD16/CD32, eBioscience) dissolved in FACS blocking buffer for 30 min on ice. Following a washing step with PBS, antibody staining was performed with PE-conjugated CD86 antibody or APC-conjugated CD206 (Thermo Scientific) for 45 min on ice. After washing in PBS, the cell pellet was resuspended in 1% PFA in PBS. Subsequently, PFA was removed, and cells were resuspended in FACS buffer and reading was taken in BD FACSVerse instrument. For flow cytometry studies in mice tissues, the harvested liver or spleen was homogenized into single-cell suspension post RBC lysis (RBC lysis Buffer, Sigma). The homogenized cell suspension was washed and resuspended in FACS buffer containing the PE-conjugated rat anti-mouse F4/80 antibody (BD Horizon, Cat# 565411) or eFluor450-conjugated anti-mouse CD11c antibody (eBioscience), or FITC-conjugated anti-mouse Ly6C antibody (BD Pharmingen) or Alexa 647-conjugated anti-mouse CD45 antibody (BioLegend). Following staining protocol, the cells were washed in PBS and the cells were resuspended in FACS buffer and the FACS protocol was performed either in BD FACSVerse or Cyto-FLEX flow cytometer (Beckman).

For DCFDA staining, 1 hr before the indicated time point of infection, cells were incubated with 10 µM DCFDA containing DMEM media at 37°C humidified incubator with 5% $CO_2$ for 45 min. Post incubation, cells were washed and harvested in PBS. Readings were measured in BD FACSVerse instrument.

All analyses were done using BD FACSuite software.

## Phenol red-hydrogen peroxidase assay

Post 48 hr of transfection, cells were infected with STM culture at an MOI of 10. Cells were incubated with phenol red solution containing hydrogen peroxidase enzyme (8.5 U/ml). At the designated time points post-infection, the supernatant was collected, and the absorbance was taken at 610 nm in Tecan Plate reader. The exogenously produced $H_2O_2$ was quantified using a standard curve of known concentration of $H_2O_2$.

## Gene expression studies by nanoString nCounter technology

Total RNA was isolated at specific time points post-infection using TRIzol (Takara) as per the manufacturer's protocol. Quantification of RNA was performed in NanoDrop (Thermo Fisher Scientific) and Qubit Bioanalyzer (Agilent 2100 Bioanalyzer). Quality of isolated RNA was detected by performing 2% agarose gel electrophoresis. Post quality check, samples were subjected to nanoString nCounter technology (theraCUES). This technology allows multi-plex, spatially resolved RNA expression quantification with appropriate probes designed against the target gene.

## Lactate estimation assay

Cell supernatant was harvested at the specific time-points post-infection, and the lactate content of the sample was estimated using the Lactate Assay Kit (Sigma, Cat# MAK064) as per the manufacturer's protocol. Briefly, 50 µl of the sample was added to each 96-well plate and each of the well 50 µl of the master reaction mix containing 46 µl of lactate assay buffer, 2 µl of lactate enzyme mix, and 2 µl of lactate probe was added. After the addition of the master reaction mix to the sample, they were mixed by horizontal shaker or via pipetting. The plate was incubated for 30 min in dark at RT. Post incubation, absorbance was measured at 570 nm. The lactate content of the sample was estimated from the lactate standard curve ranging from 0 to 10 nmole/µl.

## Fatty acid oxidation assay

At 16 hr post-infection, cell pellets were harvested and stored at –80°C. The FAO assay protocol was followed as per the manufacturer's instructions (AssayGenie Fatty Acid Oxidation [FAO] Assay Kit, Catalogue code BR00001). Briefly, cell pellets were lysed using 1X Cell lysis buffer (provided in the kit), and the cell supernatant was obtained post centrifugation of the cell lysate in a cold microfuge at 14,000 rpm for 5 min. The protein content of the cell supernatants was estimated using the Bradford (Bio-Rad) method. 20 µl of the protein sample was added to each 96-well plate in duplicate on ice. Each sample was treated with 50 µl of control solution and 50 µl of reaction solution by swiftly adding one 50 µl of control solution to one set of wells and 50 µl of reaction solution to the other set of wells. The contents were gently mixed for 10 s. The plate is covered and incubated in a 37°C incubator for 30–60 min (without $CO_2$). After incubation, cherry red colour appears in the wells. The O.D. is measured at 492 nm using a plate reader at 30 min, 60 min or 120 min. The control well reading was subtracted from the reaction well reading for each sample for each time point. The subtracted reading is used for enzyme activity calculation by considering the incubation time.

## Animal experiment

For all experiments, 6- to 8-week-old adult male C57BL/6 or *cybb-/-* mice were used. For organ burden analysis, 6-week-old C57BL/6 or *cybb-/-* mice were infected with $10^7$ CFU bacteria via oral gavaging. For bacterial enumeration via flow cytometry, mice were infected with $10^7$ GFP-expressing bacteria orally. Infected mice were intraperitoneally injected on every other day with either 1 mg/kg body weight of SIRT1 inhibitor EX-527 (Sigma-Aldrich) (*Morató et al., 2022*), or SIRT3 inhibitor 3TYP (Selleck Chemical) (*Hu et al., 2022*), or SIRT1 activator SRT1720 (Calbiochem, Sigma-Aldrich) (*Tan et al., 2021*), or treated with vehicle alone. Then, 5 days post-infection, mice were sacrificed, and bacterial organ load was estimated by plating the tissue homogenates on SS agar plates. For calculating percentage survival, 6-week-old C57BL/6 mice were infected with $10^8$ bacteria orally and monitored till fatality. For flow cytometry studies, the harvested liver or spleen were homogenized into single-cell suspension and were subjected to flow cytometry. The animal experiments were carried out in accordance with the approved guidelines of the institutional animal ethics committee at the Indian Institute of Science, Bangalore, India (registration no.: 48/1999/CPCSEA). All procedures involving

the use of animals were performed according to the Institutional Animal Ethics Committee-approved protocol.

## In vivo knockdown

For in vivo knockdown, adeno-associated virus serotype 6 (AAV6) was used. AAV6 viruses were produced in HEK293T cells. Briefly, HEK293T cells were transfected with AAV plasmid encoding shRNAs targeting SIRT1, SIRT3, and scramble control under U6 promoter together with helper plasmid using PEI Max 40000 (Polysciences, USA). Next day, transfection medium was removed, and fresh culture medium was added. Then, 48 hr later, medium was collected, and fresh culture medium was added and incubated for the next 48 hr. Again, culture medium was collected and cells were harvested by trypsinization. Collected medium containing secreted AAV viral particles was incubated with 40% PEG 8000 overnight. Precipitated viral particles were then collected by centrifugation and re-suspended in PBS. Cells containing AAV viral particles were then lysed with citrate buffer and incubated with 40% PEG 8000 and processed similarly to medium for viral particle precipitation. The precipitated viral particles were then cleaned with chloroform and loaded on iodixanol gradient for further cleaning using ultracentrifugation. Cleaned viral particles were collected from the 40% gradient and loaded on Amicon 100K cut-off columns. Purified AAV particles were then titrated by real-time PCR. For infection of AAV6, the mice were intravenously injected with a volume of 200 µl containing approximately $10^{12}$ viral particles harbouring the respective scrambled or shRNA constructs. The mice were infected on seventh day after injection of the virus with $10^6$ CFU units of *S.* Typhimurium orally. Fifth day post-infection, mice were euthanized and dissected out for organ harvesting and blood collection. The knockdown was validated by performing western blotting and qPCR of the harvested liver tissue (***Figure 10—figure supplement 1***).

## Haematoxylin and eosin staining

Here, 6- to 8-week-old C57BL/6 mice were infected with $10^7$ bacteria orally. Infected mice were intra-peritoneally injected every alternate with either 1 mg/kg body weight of SIRT1 inhibitor EX-527, or SIRT3 inhibitor 3TYP or SIRT1 activator SRT1720 or treated with vehicle alone. 5 days post-infection, mice were euthanized, and livers were collected and fixed using 3.5% paraformaldehyde. The fixed liver was then dehydrated using a gradually increasing concentration of ethanol and embedded in paraffin. 5 µm sections were collected on coated plates. Sections were further rehydrated and then stained with haematoxylin and eosin. Images were collected in a Leica microscope. **Scoring system:** according to pathological changes, the tissue sections are scored as 0 for normal pathology, 1 for mild/minor pathology, 2 for moderate pathology, and 3 for severe pathological changes.

## Statistical analysis

Data were analyzed and graphed using the GraphPad Prism 8 software (San Diego, CA). Statistical significance was determined by Student's *t*-test or two-way ANOVA and Bonferroni post-*t*-test to obtain p-values. ANOVA was used during comparison among three or more groups whereas Student's *t*-test was used to compare the difference between two groups. Mann –Whitney *t*-test was used as a non-parametric counterpart of Student's *t*-test when variances differed. Adjusted p-values<0.05 are considered statistically significant. The results are expressed as mean ± SD or SEM of three independent experiments.

## Materials availability statement

Materials are available upon request and availability.

## Acknowledgements

We thank Prof. Subba Rao Gangi Setty and Prof. Michael Hensel for providing us with the shRNA knockdown constructs and the *S.* Typhimurium 14028S strain, respectively. This work was supported by the DAE SRC fellowship (DAE00195) and DBT-IISc partnership umbrella program for advanced research in biological sciences and Bioengineering to DC. Infrastructure support from ICMR (Centre for Advanced Study in Molecular Medicine), DST (FIST), and UGC (special assistance) is highly acknowledged. DH sincerely acknowledges the CSIR-SPM fellowship for her financial support. SKG is supported by Ramalingaswami Re-entry Fellowship BT/RLF/re-entry/14/2019 from DBT, Government

of India. The funders had no role in the study design, data collection and analysis, decision to publish, or preparation of the manuscript.

## Additional information

### Funding

| Funder | Grant reference number | Author |
|---|---|---|
| Council of Scientific and Industrial Research, India | SPM-07/079(0293)/2019-EMR-I (CSIR Shyama Prasad Mukherjee Fellowship) | Dipasree Hajra |
| Department of Atomic Energy, Government of India | DAE00195 | Dipshikha Chakravortty |
| Department of Biotechnology, Ministry of Science & Technology, India | DBT-IISC | Dipshikha Chakravortty |
| Indian Council of Medical Research | | Dipshikha Chakravortty |
| Department of Science & Technology, Ministry of Science and Technology, India | | Dipshikha Chakravortty |
| Department of Biotechnology, Ministry of Science and Technology, India | BT/RLF/re-entry/14/2019 | Shashi Kumar Gupta |

The funders had no role in study design, data collection and interpretation, or the decision to submit the work for publication.

### Author contributions
Dipasree Hajra, Conceptualization, Resources, Data curation, Software, Formal analysis, Funding acquisition, Validation, Investigation, Visualization, Methodology, Writing – original draft, Writing – review and editing; Raju S Rajmani, Ayushi Devendrasingh Chaudhary, Methodology, Writing – review and editing; Shashi Kumar Gupta, Methodology, Project administration, Writing – review and editing; Dipshikha Chakravortty, Supervision, Funding acquisition, Project administration, Writing – review and editing

### Author ORCIDs
Dipasree Hajra ⬤ https://orcid.org/0000-0002-4970-8638
Dipshikha Chakravortty ⬤ https://orcid.org/0000-0002-7838-5145

### Ethics
The animal experiments were carried out in accordance with the approved guidelines of the institutional animal ethics committee at the Indian Institute of Science, Bangalore, India (Registration No: 48/1999/CPCSEA). All procedures involving the use of animals were performed according to the Institutional Animal Ethics Committee (IAEC)-approved protocol.

Reviewer #2 (Public review): https://doi.org/10.7554/eLife.93125.4.sa1
Reviewer #3 (Public review): https://doi.org/10.7554/eLife.93125.4.sa2
Author response https://doi.org/10.7554/eLife.93125.4.sa3

## Additional files

### Supplementary files
• MDAR checklist

### Data availability
All data generated or analyzed during the study are included in the manuscript and supporting files. The source data files for main and supplementary figures have been provided.

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
