## [Editor Report · eLife Assessment]

These authors present findings on the role of the sirtuins SIRT1 and SIRT3 during *Salmonella* Typhimurium infection. This **valuable** study increases our understanding of the mechanisms used by this pathogen to interact with its host and may have implications for other intracellular pathogens. The reviewers disagreed on the strength of the evidence to support the claims. Although one reviewer found the strength of the evidence **convincing**, the other found that it was **incomplete**, and that the main claims are only partially supported, as can be seen from the public reviews.

---

## [Referee Report · Reviewer #2 (Public review)]

Dipasree Hajra et al demonstrated that *Salmonella* was able to modulate the expression of Sirtuins (Sirt1 and Sirt3) and regulate the metabolic switch in both host and *Salmonella*, promoting its pathogenesis. The authors found *Salmonella* infection induced high levels of Sirt1 and Sirt3 in macrophages, which were skewed toward the M2 phenotype allowing *Salmonella* to hyper-proliferate. Mechanistically, Sirt1 and Sirt3 regulated the acetylation of HIF-1alpha and PDHA1, therefore mediating *Salmonella-induced* host metabolic shift in the infected macrophages. Interestingly, Sirt1 and Sirt3-driven host metabolic switch also had an effect on the metabolic profile of *Salmonella*. Counterintuitively, inhibition of Sirt1/3 led to increased pathogen burdens in an in vivo mouse model. Overall, this is a well-designed study.

The revised manuscript has addressed all of the previous comments. The re-analysis of flow cytometry and WB data by authors makes the results and conclusion more complete and convincing.

---

## [Referee Report · Reviewer #3 (Public review)]

Summary:

In this paper Hajra et al have attempted to identify the role of Sirt1 and Sirt3 in regulating metabolic reprogramming and macrophage host defense. They have performed gene knock down experiments in RAW macrophage cell line to show that depletion of Sirt1 or Sirt3 enhances the ability of macrophages to eliminate *Salmonella* Typhimurium. However, in mice inhibition of Sirt1 resulted in dissemination of the bacteria but the bacterial burden was still reduced in macrophages. They suggest that the effect they have observed is due to increased inflammation and ROS production by macrophages. They also try to establish a weak link with metabolism. They present data to show that the switch in metabolism from glycolysis to fatty acid oxidation is regulated by acetylation of Hif1a, and PDHA1.

Strengths:

The strength of the manuscript is that the role of Sirtuins in host-pathogen interactions have not been previously explored in-depth making the study interesting. It is also interesting to see that depletion of either Sirt1 or Sirt3 result in a similar outcome.

Weaknesses:

The major weakness of the paper is the low quality of data, making it harder to substantiate the claims. Also, there are too many pathways and mechanisms being investigated. It would have been better if the authors had focussed on either Sirt1 or Sirt3 and elucidated how it reprograms metabolism to eventually modulate host response against *Salmonella* Typhimurium. Experimental evidences are also lacking to prove the proposed mechanisms. For instance they show correlative data that knock down of Sirt1 mediated shift in metabolism is due to HIF1a acetylation but this needs to be proven with further experiments.

---

## [Author Response]

The following is the authors’ response to the current reviews.

**Reviewer #2 (Public Review):**
Dipasree Hajra et al demonstrated that Salmonella was able to modulate the expression of Sirtuins (Sirt1 and Sirt3) and regulate the metabolic switch in both host and Salmonella, promoting its pathogenesis. The authors found Salmonella infection induced high levels of Sirt1 and Sirt3 in macrophages, which were skewed toward the M2 phenotype allowing Salmonella to hyper-proliferate. Mechanistically, Sirt1 and Sirt3 regulated the acetylation of HIF-1alpha and PDHA1, therefore mediating Salmonella-induced host metabolic shift in the infected macrophages. Interestingly, Sirt1 and Sirt3-driven host metabolic switch also had an effect on the metabolic profile of Salmonella. Counterintuitively, inhibition of Sirt1/3 led to increased pathogen burdens in an in vivo mouse model. Overall, this is a well-designed study.The revised manuscript has addressed all of the previous comments. The re-analysis of flow cytometry and WB data by authors makes the results and conclusion more complete and convincing.

We are immensely grateful to the reviewer for improving the strength of the manuscript by providing insightful comments and for appreciating the work.

**Reviewer #3 (Public Review):**
Summary:In this paper Hajra et al have attempted to identify the role of Sirt1 and Sirt3 in regulating metabolic reprogramming and macrophage host defense. They have performed gene knock down experiments in RAW macrophage cell line to show that depletion of Sirt1 or Sirt3 enhances the ability of macrophages to eliminate Salmonella Typhimurium. However, in mice inhibition of Sirt1 resulted in dissemination of the bacteria but the bacterial burden was still reduced in macrophages. They suggest that the effect they have observed is due to increased inflammation and ROS production by macrophages. They also try to establish a weak link with metabolism. They present data to show that the switch in metabolism from glycolysis to fatty acid oxidation is regulated by acetylation of Hif1a, and PDHA1.Strengths:The strength of the manuscript is that the role of Sirtuins in host-pathogen interactions have not been previously explored in-depth making the study interesting. It is also interesting to see that depletion of either Sirt1 or Sirt3 result in a similar outcome.Weaknesses:The major weakness of the paper is the low quality of data, making it harder to substantiate the claims. Also, there are too many pathways and mechanisms being investigated. It would have been better if the authors had focussed on either Sirt1 or Sirt3 and elucidated how it reprograms metabolism to eventually modulate host response against Salmonella Typhimurium. Experimental evidences are also lacking to prove the proposed mechanisms. For instance they show correlative data that knockdown of Sirt1 mediated shift in metabolism is due to HIF1a acetylation but this needs to be proven with further experiments.

As the public review of the reviewer remains unaltered as the previous version without further recommendations for authors, we are sticking to our former author’s response. We respect the reviewer’s opinion and thank the reviewer for the critical analysis of our work.

--------

The following is the authors’ response to the previous reviews.

**Reviewer #2 (Public Review):**
Dipasree Hajra et al demonstrated that Salmonella was able to modulate the expression of Sirtuins (Sirt1 and Sirt3) and regulate the metabolic switch in both host and *Salmonella*, promoting its pathogenesis. The authors found Salmonella infection induced high levels of Sirt1 and Sirt3 in macrophages, which were skewed toward the M2 phenotype allowing Salmonella to hyper-proliferate. Mechanistically, Sirt1 and Sirt3 regulated the acetylation of HIF-1alpha and PDHA1, therefore mediating *Salmonella-induced* host metabolic shift in the infected macrophages. Interestingly, Sirt1 and Sirt3-driven host metabolic switch also had an effect on the metabolic profile of Salmonella. Counterintuitively, inhibition of Sirt1/3 led to increased pathogen burdens in an in vivo mouse model. Overall, this is a well-designed study.Comments on revised version:The authors have performed additional experiments to address the discrepancy between in vitro and in vivo data. While this offers some potential insights into the in vivo role of Sirt1/3 in different cell types and how this affects bacterial growth/dissemination, I still believe that Sirt1/3 inhibitors could have some effect on the gut microbiota contributing to increased pathogen counts. This possibility can be discussed briefly to give a better scenario of how Sirt1/3 inhibitors work in vivo. Additionally, the manuscript would improve significantly if some of the flow cytometry analysis and WB data could be better analyzed.

We are highly grateful for your valuable and insightful comments. Thank you for appreciating the merit of our manuscript. As rightly pointed out by the eminent reviewer, we acknowledge the probable link of Sirtuin on gut microbiota and its effect on increased bacterial loads as indicated by previous literature studies (PMID: 22115311, PMID: 19228061). These reports suggested that a low dose of Sirt1 activator, resveratrol treatment in rats for 25 days treatment under 5% DSS induced colitis condition led to alterations in gut microbiota profile with increased lactobacilli and bifidobacteria alongside reduced abundance of enterobacteria. This study correlates with our study wherein we have detected enhanced *Salmonella* (belonging to Enterobacteriaceae family) loads under both Sirt1/3 in vivo knockdown condition or inhibitor-treated condition in C57BL/6 mice and reduced burden under Sirt-1 activator treatment SRT1720.

As per your valid suggestion, we have discussed this possibility in our discussion section. (Line- 541-548).

We have incorporated the suggestions for the improvement in the analysis of WB data and flow cytometry.

**Reviewer #3 (Public Review):**
Summary:In this paper Hajra et al have attempted to identify the role of Sirt1 and Sirt3 in regulating metabolic reprogramming and macrophage host defense. They have performed gene knock down experiments in RAW macrophage cell line to show that depletion of Sirt1 or Sirt3 enhances the ability of macrophages to eliminate *Salmonella* Typhimurium. However, in mice inhibition of Sirt1 resulted in dissemination of the bacteria but the bacterial burden was still reduced in macrophages. They suggest that the effect they have observed is due to increased inflammation and ROS production by macrophages. They also try to establish a weak link with metabolism. They present data to show that the switch in metabolism from glycolysis to fatty acid oxidation is regulated by acetylation of Hif1a, and PDHA1.Strengths:The strength of the manuscript is that the role of Sirtuins in host-pathogen interactions has not been previously explored in-depth making the study interesting. It is also interesting to see that depletion of either Sirt1 or Sirt3 results in a similar outcome.Weaknesses:The major weakness of the paper is the low quality of data, making it harder to substantiate the claims. Also, there are too many pathways and mechanisms being investigated. It would have been better if the authors had focussed on either Sirt1 or Sirt3 and elucidated how it reprograms metabolism to eventually modulate host response against *Salmonella* Typhimurium. Experimental evidence is also lacking to prove the proposed mechanisms. For instance they show correlative data that knock down of Sirt1 mediated shift in metabolism is due to HIF1a acetylation but this needs to be proven with further experiments.

We appreciate the reviewer’s critical analysis of our work. In the revised manuscript, we aimed to eliminate the low-quality data sets and have tried to substantiate them with better and conclusive ones, as directed in the recommendations for the author section. We agree with the reviewer that the inclusion of both Sirtuins 1 and 3 has resulted in too many pathways and mechanisms and focusing on one SIRT and its mechanism of metabolic reprogramming and immune modulation would have been a less complicated alternative approach. However, as rightly pointed out, our work demonstrated the shared and few overlapping roles of the two sirtuins, SIRT1 and SIRT3, together mediating the immune-metabolic switch upon *Salmonella* infection. As per the reviewer’s suggestion, we have performed additional experiments with HIF-1α inhibitor treatment in our revised manuscript to substantiate our correlative findings on SIRT1-mediated regulation of host glycolysis (Fig.7G). We wanted to clarify our claim in this regard. Our results suggested that loss of SIRT1 function triggered increased host glycolysis alongside hyperacetylation of HIF-1α. HIF-1α is reported to be one of the important players in glycolysis regulation (Kierans SJ, Taylor CT. Regulation of glycolysis by the hypoxia-inducible factor (HIF): implications for cellular physiology. J Physiol. 2021;599(1):23-37. doi:10.1113/JP280572.) and additionally, SIRT1 has been shown to regulate HIF-1α acetylation status (Lim JH, Lee YM, Chun YS, Chen J, Kim JE, Park JW. Sirtuin 1 modulates cellular responses to hypoxia by deacetylating hypoxia-inducible factor 1 alpha. Mol Cell. 2010;38(6):864-878. doi:10.1016/j.molcel.2010.05.023.) Further, ectopic expression of SIRT1 has been demonstrated to reduce glycolysis by negatively regulating HIF-1α. (Wang Y, Bi Y, Chen X, et al. Histone Deacetylase SIRT1 Negatively Regulates the Differentiation of Interleukin-9-Producing CD4(+) T Cells. Immunity. 2016;44(6):1337-1349. doi:10.1016/j.immuni.2016.05.009). We have subsequently shown in Fig. 7G, that the increase in host glycolysis upon SIRT knockdown in the infected macrophages gets lowered upon HIF-1α inhibitor treatment, suggesting that one of the mechanisms of SIRT-mediated regulation of host glycolysis is via regulation of HIF-1α. However, this warrants further future mechanistic research.

**Recommendations for the authors:**

**Reviewer #2 (Recommendations For The Authors):**
(1) Figures 8I-S: are only viable cells used for analysis? Please provide gating strategy used for these analyses.(2) Many changes seen in WB seem to be marginal. Since the authors used densitometric plot to quantify the band intensities, I expect these experiments were repeated at least three times. Please indicate the number of repeats. For instance, Figures 7C, 7I (UI SCR vs UI shSIRT3), 7J, show marginal changes or no changes. What do other WB images look like? Are they more convincing than the ones currently shown? Please provide them in the response letter.(3) Figure 7C: label is a bit misleading. Please relabel the figure title to Acetylated HIF vs total levels(4) Figure 7J: which band is AcPDHA1?

(1) We are highly apologetic for not clarifying our gating strategy for the analysis.

We initially gated the viable splenocyte population based on Forward scatter (FSC) and Side Scatter (SSC). This gated population was further subjected to gating based on cell FSC-H (height) versus FSC-A (area). Subsequently, the population was gated as per SSC-A and GFP (expressed by intracellular bacteria) based on the autofluorescence exhibited by the uninfected control (Fig. 8I-J).

**Author response image 1. sa3fig1:** Uninfected.

**Author response image 2. sa3fig2:** Vehicle control infected.

**Author response image 3. sa3fig3:** EX-527 infected.

**Author response image 4. sa3fig4:** 3TYP infected.

**Author response image 5. sa3fig5:** SRT 1720 infected.

For gating different cell types such as F4/80 (PE) positive population in Fig. 8K-L, the viable cell population was gated based on SSC-A versus PE-A to gate the macrophage population. These macrophage populations were gated further based on GFP (*Salmonella*) + population to obtain the percentage of macrophage population harboring GFP+ bacteria. Similar strategies were followed for other cell types as depicted in Fig. 8M-S, Fig. S8.

(2) We agree with the reviewer’s concern with the marginal changes in the western blots (Figures 7C, 7I (UI SCR vs UI shSIRT3), 7J). As per the suggestions, we have provided the alternate blot images and have indicated the number of repeats in the manuscript. The alternate blot images are provided herewith:

**Author response image 6. sa3fig6:** Alternate blot images for Fig. 7B-C.

**Author response image 7. sa3fig7:** Alternate blot images for Fig. 7I, J.

(1) We are highly thankful to the reviewer for recommending this suggestion. We have made the necessary modifications of relabelling Fig. C to Acetylated HIF-1α over total HIF-1α as per the suggestion.

(2) 7J Acetylated PDHA1 has been duly pointed as per the suggestion. We are extremely apologetic for the inconvenience caused.

**Author response image 8. sa3fig8:** 

**Reviewer #3 (Recommendations For The Authors):**
The authors have done some work to improve the manuscript. However, the data presented lacks clarity.Fig 4B: I still do not see a change in Ac p65 in the less saturated blot. It looks reduced as the band is distorted. I am not sure how this could be quantified.Fig S2 b-actin bands are hyper saturated, and it is not possible to decipher the knockdown efficiency. It is probably better to provide a ponceau staining similar to S2C. The band intensity values are out of place.Fig 5F HADHA blot: Lane 1 expression appears to be significantly higher than lane 3, but the values mentioned do not match the intensity of the bands.It is hard to interpret the authors' claim that the shift in metabolism is HIF1a-dependent.Fig 7B: I would expect HIF1a acetylation to be increased in UI ShSIRT1 compared to UI SCR. The blot shows reduced HIF1a acetylation.Fig 7D: SIRT1 immunoprecipitates with HIF1a equally under all conditions. Is this what the authors expect? Labelling of the blots are not clear. It looks like the bottom SIRT1 blot is from Beads IgG control.Fig 7H: How does PDHA1 interact with SIRT3 so strongly in shSIRT3 cells (lane 2)?Authors have mentioned in their response that a knockdown of 40% has been achieved in the uninfected but the blot does not reflect that. SIRT3 expression seems to be more in the knockdown.Blots are also not labelled properly especially Input. The lanes are not marked.

We thank the reviewer for acknowledging the improvements in the revised version and for suggesting further clarifications and improvements.

We have tried to incorporate the specified modifications to the best of our abilities in the revised manuscript.

We are highly apologetic for the inconclusive blot image in the figure 4B. We have provided an alternative blot image with better clarity for Fig.4B used for quantification analysis.

**Author response image 9. sa3fig9:** 

As per the reviewer’s valuable suggestions, we have provided the ponceau image in the Fig. S2B.

We thank the reviewers for rightly pointing out the discrepancy in the band intensity quantification in the Fig. 5F. We have re-evaluated the intensities on imageJ and have provided with the correct band intensities. We are highly apologetic for the inaccuracies.

As per the reviewer’s previous suggestion, we have performed additional experiments with HIF-1α inhibitor treatment in our revised manuscript to substantiate our correlative findings on SIRT1-mediated regulation of host glycolysis (Fig.7G). We wanted to clarify our claim in this regard. Our results suggested that loss of SIRT1 function triggered increased host glycolysis alongside hyperacetylation of HIF-1α. HIF-1α is reported to be one of the important players of glycolysis regulation (Kierans SJ, Taylor CT. Regulation of glycolysis by the hypoxia-inducible factor (HIF): implications for cellular physiology. J Physiol. 2021;599(1):23-37. doi:10.1113/JP280572.) and additionally, SIRT1 has been shown to regulate HIF-1α acetylation status (Lim JH, Lee YM, Chun YS, Chen J, Kim JE, Park JW. Sirtuin 1 modulates cellular responses to hypoxia by deacetylating hypoxia-inducible factor 1alpha. Mol Cell. 2010;38(6):864-878. doi:10.1016/j.molcel.2010.05.023.) Further, ectopic expression of SIRT1 has been demonstrated to reduce glycolysis by negatively regulating HIF-1α. (Wang Y, Bi Y, Chen X, et al. Histone Deacetylase SIRT1 Negatively Regulates the Differentiation of Interleukin-9-Producing CD4(+) T Cells. Immunity. 2016;44(6):1337-1349. doi:10.1016/j.immuni.2016.05.009). We have subsequently shown in Fig. 7G, that the increase in host glycolysis upon SIRT knockdown in the infected macrophages gets lowered upon HIF-1α inhibitor treatment, suggesting that one of the mechanisms of SIRT-mediated regulation of host glycolysis is via regulation of HIF-1α. However, this warrants further future mechanistic research.

We agree with the reviewer’s claim of increased HIF-1α acetylation in the UI sh1 versus UI SCR. The apparent reduced acetylation depicted in UI sh1 in Fig. 7B could be attributed to lower HIF-1α levels in the UI sh1 compared to UI SCR. Therefore, we have provided an alternate blot image that been used for quantification in Fig. 7C (Author response image 6).

To answer the reviewer’s question in Fig. 7D, we have noticed more or less equal degree of immunoprecipitation of HIF-1α under pull down of HIF-1α in all the sample cohorts under conditions of SIRT1 inhibitor treatment. However, we have observed reduced interaction of HIF-1α with SIRT1 in the infected sample upon SIRT1 inhibitor treatment.

We thank the reviewers for suggesting improvements in the blot labelling and for raising this concern. We have corrected the blot labelling to avoid the previous confusion.

We appreciate the reviewer’s concern and therefore we have provided an alternate blot image for Fig. 7H which might address the previous stated concern wherein we have achieved an enhanced SIRT3 knockdown percentage.

We are extremely apologetic for the improper labelling of the Input blot with unmarked lanes. We have addressed this issue by labelling the lanes in the input section of the blots.